# Top-$k$ Feature Importance Ranking

**Yuxi Chen**  *ericc3@andrew.cmu.edu*
*Carnegie Mellon University*

**Tiffany Tang**  *ttang4@nd.edu*
*University of Notre Dame*

**Genevera Allen**  *genevera.allen@columbia.edu*
*Columbia University*

**Reviewed on OpenReview:** *https://openreview.net/forum?id=2OSHpccsaV*

## Abstract

Accurate ranking of important features is a fundamental challenge in interpretable machine learning with critical applications in scientific discovery and decision-making. Unlike feature selection and feature importance, the specific problem of ranking important features has received considerably less attention. We introduce RAMPART (Ranked Attributions with MiniPatches And Recursive Trimming), a framework that utilizes any existing feature importance measure in a novel algorithm specifically tailored for ranking the top-$k$ features. Our approach combines an adaptive sequential halving strategy that progressively focuses computational resources on promising features with an efficient ensembling technique using both observation and feature subsampling. Unlike existing methods that convert importance scores to ranks as post-processing, our framework explicitly optimizes for ranking accuracy. We provide theoretical guarantees showing that RAMPART achieves the correct top-$k$ ranking with high probability under mild conditions, and demonstrate through extensive simulation studies that RAMPART consistently outperforms popular feature importance methods, concluding with two high-dimensional genomics case studies. Our code is available at `https://github.com/DataSlingers/TopK`.

## 1 Introduction

A key challenge in interpretable machine learning is determining not just which features influence model predictions but their relative importance ranking. In particular, accurately ranking the top-$k$ most important features would fundamentally change the decision-making and scientific discovery process in numerous high-stakes applications (Bhatt et al., 2020; Jaxa-Rozen and Trutnevyte, 2021). For example, in many social science surveys, there are far more questions that could, in principle, be asked than respondents can answer without incurring fatigue or disengagement. Researchers therefore often conduct small pilot studies with a larger question bank to identify the most important questions to retain in the final survey instrument (Jeong et al., 2023; DeVellis, 2017). In genomics, genome-wide association studies (GWAS) (Visscher et al., 2017) are by far the most common approach to identify important genes or genetic variants that are associated with disease risk. These data-driven studies often identify hundreds of important genetic variants. However, to translate these findings into tangible therapeutic targets and clinical practice, wet-lab validation is necessary, but typically limited to a few dozen genetic variants, if not fewer due to its high cost (Fu et al., 2020; Wang et al., 2025; Pashaei et al., 2025). More generally, given resource constraints, the need to rank or prioritize a small number of top-ranked candidates for costly downstream decision-making is a common theme among many clinical and scientific pipelines.

Although feature importances have been extensively studied in machine learning, methods specifically designed for ranking the top-$k$ most important features remain underdeveloped. Current approaches for top-$k$ feature

ranking typically rely on the heuristic of first estimating feature importance values for all features, sorting them, and then subsetting to the top-$k$ features with the largest importance (Lundberg and Lee, 2017; Neuhof and Benjamini, 2024; Goldwasser and Hooker, 2025). However, the first step of this paradigm is particularly limiting as valuable data and computational resources are being used to estimate the importances of *all* features, including those that are irrelevant or far outside of the top-$k$ that are of primary interest. This issue is further exacerbated in realistic settings with correlated and high-dimensional data (e.g., in genomics), where existing feature importance estimates are known to be highly unstable and unreliable (Nicodemus and Malley, 2009; Nicodemus, 2011; Hooker et al., 2021). These challenges highlight the need for a paradigm that directly targets the top-$k$ ranking task, which focuses computational and statistical resources on the most important features and can effectively handle correlated, high-dimensional data.

## 1.1 Our Contributions

Motivated by these challenges, we focus on the problem of directly ranking the top-$k$ features with the highest global importance and develop a model-agnostic framework tailored to this task. To effectively handle correlated features, we first introduce RAMP (Ranked Attributions with MiniPatches), an ensembling strategy that aggregates feature importances from models trained on random subsamples (or "minipatches") of both observations and features. This minipatch-based approach breaks harmful correlation patterns among features while maintaining statistical power (Gan et al., 2022). Building on RAMP, we then develop RAMPART (RAMP And Recursive Trimming), which leverages an adaptive strategy to progressively focus computational resources on promising features while eliminating suboptimal ones. Unlike existing approaches that allocate equal resources to all features, this adaptive strategy enables RAMPART to grow increasingly precise in distinguishing between similarly-ranked top features as the candidate pool shrinks. Importantly, both RAMP and RAMPART are model-agnostic frameworks for top-$k$ feature importance ranking that can serve as wrappers around any existing feature attribution method. Finally, we provide theoretical guarantees on recovering the correct top-$k$ feature importance ranking under mild assumptions, establishing explicit sample complexity bounds that may be of independent interest.

## 1.2 Related Works

**Feature Importance** Although not directly designed for feature importance ranking, numerous model-specific and model-agnostic feature importance measures have been developed to quantify the contribution of each predictor feature on the model's predictions and performance (Molnar, 2025). Popular model-specific approaches include regression coefficients for linear models, Mean Decrease in Impurity for tree-based methods (Breiman, 2001), and neural network attributions like DeepLift and Integrated Gradients (Shrikumar et al., 2019; Sundararajan et al., 2017). Model-agnostic methods include occlusion-based (Lei and Wasserman, 2014), permutation-based (Breiman, 2001), and Shapley-based techniques (Lundberg and Lee, 2017; Lundberg et al., 2020). In Section 4, we will demonstrate the shortcomings of simply ranking these feature importances to obtain the top-$k$.

**Ranking from Pairwise Comparisons** On the other hand, viewing this problem from the lens of the ranking literature, many previous works have directly estimated rankings from pairwise comparisons. This literature includes tournament methods (Mohajer et al., 2017), spectral techniques (Negahban et al., 2017; Chen and Suh, 2015; Chen et al., 2019), adaptive selection paradigms (Heckel et al., 2016; 2018), and weighting strategies (Shah and Wainwright, 2018; Wauthier et al., 2013; Ammar and Shah, 2012). Despite their theoretical appeal, these methods struggle to capture multivariate feature dependencies and face computational barriers in high dimensions, limiting their applicability to top-$k$ feature importance ranking.

**Feature Importance Ranking** More recently, several works have focused on feature importance ranking specifically. Kariyappa et al. (2023) developed sampling algorithms to identify top-$k$ features by Shapley values without addressing their ordering. Teneggi and Sulam (2024) employed statistical independence testing with betting principles, primarily for semantic concept validation in vision rather than tabular data. Neuhof and Benjamini (2024) introduced a framework for quantifying uncertainty of feature importance rankings through simultaneous confidence intervals. Their approach focuses primarily on post-hoc interpretation of

pre-computed importance scores rather than providing an efficient algorithmic framework for large-scale feature ranking. Model-free, dependence-based ordering offers an orthogonal approach to ranking variables via a nonparametric coefficient of conditional dependence and yields a tuning-free ranking without fitting predictive models (Azadkia and Chatterjee, 2021). A closely related approach by Azadkia and Roudaki (2025) leverages an integrated $R^2$ dependence measure and a greedy forward-selection procedure.

Most relevant to our work, Goldwasser and Hooker (2025) developed a sequential pairwise hypothesis testing framework for assessing the statistical significance of the top-$k$ most important features using resampled attribution scores. This approach, however, requires normality and independence assumptions that are rarely satisfied in practice and violated by correlated estimators. Computational overhead from repeated pairwise testing also limits scalability to high dimensions. Their subsequent rank verification method (Goldwasser et al., 2025) similarly assumes Gaussian distributions, constraining applicability to real-world data with non-Gaussian distributions and complex dependencies.

**Best Arm Identification**   To avoid the current limitations of existing feature importance ranking approaches, we introduce a recursive trimming strategy, which draws inspiration from multi-armed bandits research on best arm identification, including UCB approaches (Audibert et al., 2010; Chen et al., 2017), Thompson Sampling (Russo, 2020), and halving algorithms (Zhao et al., 2023). Particularly relevant is research on best-$k$-arm identification (Chen et al., 2008; Gao and Chen, 2015; You et al., 2023), with Liu and Ročková (2023) successfully applying Thompson Sampling to variable selection. However, direct application of bandit algorithms to feature ranking faces key challenges: (1) assumed arm independence—violated by correlated features and (2) unknown distributions of importance measures. Our work addresses these limitations with a novel approach that efficiently produces statistically robust feature rankings while accounting for feature interdependencies.

## 2  Adaptive Feature Importance Ranking

### 2.1  Problem Setup

Suppose we observe a dataset $\mathcal{D} = \{(\mathbf{x}_1, y_1) \ldots, (\mathbf{x}_N, y_N)\}$ where $\mathbf{x}_i \in \mathbb{R}^M$ and $y_i \in \mathbb{R}$ denote the features and response respectively. We assume observations are independent and identically distributed draws from an unknown joint distribution. We also assume that each feature $j \in [M] := \{1, \ldots, M\}$ possesses an inherent global feature importance $\phi_j$. There are many existing metrics that can be used to quantify $\phi_j$: see Lundberg and Lee (2017); Molnar (2025); Fisher et al. (2019) for instance. We note that the feature importance $\phi_j$ depends on the specific predictive model and importance metric used, with each method potentially defining a distinct ground truth. We define the rank of the $j$-th feature as $r_j := \sum_{i=1}^{M} \mathbf{1}\{|\phi_j| < |\phi_i|\}$ and the $j$-th best feature as $\tau_j$. Additionally, for some pre-specified $k \ll M$, we assume that the top-$k$ features do not contain any ties: $r_{\tau_j} \neq r_{\tau_{j'}}$ for any $j, j' \in [k]$ where $j \neq j'$. Our goal is to correctly estimate these top $k$ ranks such that $\hat{r}_{\tau_j} = r_{\tau_j}$ for $j \in [k]$.

### 2.2  Motivation

As mentioned previously, one naive approach to feature importance ranking is to first estimate importance scores $\{\hat{\phi}_j\}_{j=1}^{M}$ and then sort the features accordingly. For instance, in a high-dimensional regression model, one may often rank the features according to the magnitude of their standardized coefficients. However, in realistic, high-dimensional settings, this approach faces several fundamental challenges, which motivate our proposed framework.

First, obtaining accurate feature importance estimates in high dimensions is problematic. These estimates often suffer from bias due to inherent correlations in high-dimensional data, with well-known metrics performing poorly in correlated settings (Chamma et al., 2023a). The estimates also exhibit high variance and unreliability, as even classical methods such as ordinary least squares coefficients are known to become unstable when the number of features approaches or exceeds the sample size (Hastie et al., 2009). The computational burden compounds these issues: computing importances for a large number of features can be expensive, forcing popular approaches to resort to approximations. For instance, methods like Shapley

values are computationally infeasible without substantial approximations that compromise their theoretical guarantees (Mitchell et al., 2022; Ghorbani and Zou, 2019).

Second, this approach fundamentally misaligns computational and statistical resources with our objective. Our focus is not on ranking all features but only the top-$k$, a crucial distinction that should guide algorithm design. Analogous to sparse modeling in high-dimensional statistics, we expect many features to be noise and are uninterested in learning their ranks (Hastie et al., 2015). However, estimating and sorting importance metrics expends valuable statistical resources (e.g., data samples) and computational effort on all features, including noise features. This misallocation is particularly problematic when distinguishing between features of similar importance levels, where more precise estimation is needed.

For example, consider predicting credit risk using various financial and demographic features, which are often highly correlated. Lenders using machine learning models to model credit risk are required by law to disclose the principal reasons when denying an application or taking other adverse action (Consumer Financial Protection Bureau, 2011). While standard feature importance methods may struggle to distinguish between similarly important features due to correlations, our framework is designed to specifically address these challenges by focusing resources on the most relevant features and improving feature ranking accuracy in high-dimensional, correlated settings.

Motivated by these considerations, we develop a two-stage approach that combines efficient feature importance estimation with adaptive refinement. We first introduce RAMP (Ranked Attributions with MiniPatches), which leverages ensemble learning principles through random subsampling of both features and observations. We then extend this to RAMPART (RAMP And Recursive Trimming), which progressively focuses computational resources on the most promising features.

### 2.3 Ranked Attributions with MiniPatches (RAMP)

We begin by assuming access to a feature importance ranking procedure $\mathcal{M}$. This procedure $\mathcal{M} : \mathbb{R}^{n \times m} \times \mathbb{R}^n \mapsto \{0, \ldots, m-1\}$ takes as input a subset of the data (i.e., a "minipatch") $(\mathbf{X}_{I,F}, \mathbf{Y}_I)$ where $I \subseteq [N], |I| = n$ is a subsample of observations and $F \subseteq [M], |F| = m$ is a subsample of features, and returns rank estimates $\tilde{r}_j$ for features $j \in [F]$. In practice, this ranking procedure $\mathcal{M}$ typically involves: (i) fitting a predictive model to the minipatch, (ii) computing feature importance scores using a specific attribution method, and (iii) sorting these scores by magnitude to output ranks. For example, a simple ranking procedure $\mathcal{M}$ could entail fitting a linear regression model and ranking features by the magnitude of their standardized coefficients, or alternatively, fitting a decision tree and ranking features by their mean decrease in impurity (MDI).

Given $\mathcal{M}$, we formalize RAMP in Algorithm 1. The procedure operates by generating numerous minipatches, with each consisting of a different random subsample of both observations and features. For each minipatch, RAMP applies $\mathcal{M}$ (e.g., linear regression coefficients or decision tree with MDI) to obtain feature importance rank estimates $\tilde{r}_j^b$, which are then averaged across all minipatches where each feature appears. This minipatch ensembling approach reduces variance while breaking harmful correlation patterns between features (Gan et al., 2022). The final step sorts the averaged ranks $\bar{r}_j$ using order statistics $\bar{r}_{(1)}, \ldots, \bar{r}_{(M)}$ to obtain the final rankings.

Importantly, RAMP serves as a meta-algorithm that improves the accuracy of feature rankings regardless of the specific attribution method employed. More specifically, each ranking procedure $\mathcal{M}$ produces its own importance measure $\phi_j$ and hence ranks $r_j$, but ordinary estimates of these quantities typically suffer from instability in high-dimensions due to feature correlations and sampling variability (Chamma et al., 2023b; Kelodjou et al., 2024). Given the choice of ranking procedure $\mathcal{M}$, RAMP improves the estimation of the corresponding ranks $r_j$ by ensembling across diverse minipatches, effectively reducing variance while maintaining the statistical properties of the underlying importance measures (Gan et al., 2022; Yao and Allen, 2020). This approach provides more stable and accurate approximations of the true feature importance ranking, while maintaining the flexibility to accommodate any feature importance procedure. Since decisions are made based on ranks instead of raw attribution magnitudes, RAMP can ensemble heterogeneous per-minipatch ranking procedures (e.g., MDI, SHAP, Integrated Gradients) without modification, echoing the model-class view that genuinely important variables should remain important across a broad family of predictive models (Fisher et al., 2019).

---

**Algorithm 1** Ranked Attributions with MiniPatches (RAMP)

---

**Input:** Ranking procedure $\mathcal{M}$, dataset $\mathcal{D} = \{(\mathbf{x}_i, y_i)\}_{i=1}^N \equiv (\mathbf{X}, \mathbf{Y})$, number of minipatches $B$, data subsample size $n < N$, feature subsample size $m < M$

**Output:** Estimated feature ranks $\hat{r}_1, \ldots, \hat{r}_M$

1: **for** $b \in [B]$ **do**
2:      $I_b \leftarrow$ randomly subsample $n$ observations $\subset [N]$
3:      $F_b \leftarrow$ randomly subsample $m$ features $\subset [M]$
4:      $\tilde{r}_j^b \leftarrow \mathcal{M}(\mathbf{X}_{I_b, F_b}, \mathbf{Y}_{I_b})_j$ for $j \in F_b$
5: **end for**
6: For all $j \in [M]$, set $\bar{r}_j \leftarrow \dfrac{\sum_{b \in [B]: j \in F_b} \tilde{r}_j^b}{\sum_{b \in [B]} \mathbf{1}\{j \in F_b\}}$
7: **return** $\hat{r}_j = (i : \bar{r}_{(i)} = \bar{r}_j) - 1$

---

## 2.4 RAMP And Recursive Trimming (RAMPART)

Though RAMP provides an improved foundation for feature ranking, the uniform treatment of all features within RAMP is not ideal when our primary interest is in the top-$k$ ranked features. To address these limitations, we develop RAMPART (Ranked Attributions with MiniPatches And Recursive Trimming), an adaptive framework that builds upon the minipatch sampling from RAMP while incorporating ideas from the sequential halving literature. Our approach is inspired by the pioneering work of Karnin et al. (2013) on successive halving algorithms, as well as recent advances in batched sequential halving (Jun et al., 2016; Koyamada et al., 2024) for the fixed batch setting. While these methods were originally developed for best-arm identification in multi-armed bandits, we adapt their core insight of progressive resource allocation to the feature ranking context.

RAMPART operates by iteratively applying RAMP to an increasingly focused set of features. In each round, it identifies and retains the more promising half of the features while eliminating those less likely to be in the top-$k$ set. This adaptive strategy, formalized in Algorithm 2, recursively trims the feature pool size, enabling more accurate rank estimation in later rounds where fine-grained distinctions become crucial. The number of iterations is carefully chosen to ensure the final feature pool size aligns with our target $k$. By concentrating computational resources on the most relevant features, RAMPART more efficiently spends its resources, distinguishing between similarly-ranked important features. This adaptive resource allocation is a critical advantage over traditional approaches that uniformly evaluate all features since features that survive to later rounds in RAMPART are evaluated more frequently, enabling increasingly precise rank estimates where they matter most.

---

**Algorithm 2** Ranked Attributions with MiniPatches And Recursive Trimming (RAMPART)

---

**Input:** Ranking procedure $\mathcal{M}$, dataset $\mathcal{D} = \{(\mathbf{x}_i, y_i)\}_{i=1}^N$, number of minipatches $B$, data subsample size $n < N$, feature subsample size $m < M$, top features $k$

**Output:** Estimated feature ranks $\hat{r}_1, \ldots, \hat{r}_M$

1: $T \leftarrow \lfloor \log_2 M \rfloor - \lceil \log_2 k \rceil + 1$
2: $\mathcal{C}_1 \leftarrow [M]$
3: **for** $t \in \{1, \ldots, T\}$ **do**
4:      $\hat{r}_1^t, \ldots, \hat{r}_{|\mathcal{C}_t|}^t \leftarrow \text{RAMP}_{B,n,m}(\mathcal{C}_t)$
5:      $\mathcal{C}_{t+1} \leftarrow \{\hat{\tau}_1^t, \ldots, \hat{\tau}_{|\mathcal{C}_t|/2}^t\}$
6: **end for**
7: **return** $\hat{r}_j = \hat{r}_j^T$ if $j \in \mathcal{C}_T$ otherwise $\hat{r}_j = k$

---

# 3 Theoretical Analysis

In this section, we show theoretical guarantees for RAMP and RAMPART. We also prove that RAMPART achieves performance superior to that of RAMP under mild assumptions on the properties of the ranking procedure $\mathcal{M}$.

**Assumption 1. Unique Top-$k$ Ranks.** For any two features $j, j' \in [M]$ where at least one feature has true rank smaller than $k$, either $r_j > r_{j'}$ or $r_{j'} > r_j$.

While this assumption might appear restrictive at first glance, it only requires distinct ranks among the top-$k$ features of interest. Ties are permitted among features outside this set. This reflects real-world settings where we need to distinguish among the most important features but can tolerate ambiguity in the ordering of less relevant or null features.

**Assumption 2. Rank Consistency.** For any two features $j, j' \in \mathcal{S} \subseteq [M], |\mathcal{S}| \geq m$ with $r_j < r_{j'}$ that are sampled in the same minipatch,

$$\mathbf{P}\big(\tilde{r}_j < \tilde{r}_{j'}\big|j, j' \in F\big) \geq p > \frac{1}{2}$$

where the probability is taken over all minipatches $F$ of size $m$ in $\mathcal{S}$ such that $j$ and $j'$ are sampled together.

This consistency assumption requires that our ranking procedure $\mathcal{M}$ performs better than random guessing when comparing features within the same minipatch on average. If a model cannot reliably order features when evaluated together, it cannot be expected to produce meaningful relative rankings. In particular, we only require probabilistic consistency, allowing for errors in individual comparisons.

**Assumption 3. Unbiased Ordering.** For any two features $j, j' \in \mathcal{S} \subseteq [M], |\mathcal{S}| \geq m$ with $r_j < r'_j$,

$$\mathbf{E}\big[\tilde{r}_j\big|j \in F, j' \notin F\big] < \mathbf{E}\big[\tilde{r}_{j'}\big|j' \in F, j \notin F\big]$$

where expectations are taken over minipatches in $\mathcal{S}$ that sample one feature but not the other.

This assumption says that rank comparisons should remain informative across different minipatches. It requires that features of higher importance tend to receive better ranks relative to less important features, even when they appear in separate samples rather than being directly compared. This property is especially relevant for our minipatch approach, since we aggregate rank estimates across many different subsamples where not all pairs of features appear together.

**Assumption 4. Bounded Deviation.** There exists a universal constant $C > 0$ such that for any two features $j, j' \in \mathcal{S} \subseteq [M], |\mathcal{S}| \geq m$ with $r_j < r_{j'}$ that are sampled in the same minipatch,

$$\mathbf{E}[\tilde{r}_{j'} - \tilde{r}_j|\tilde{r}_{j'} > \tilde{r}_j] - \mathbf{E}[\tilde{r}_j - \tilde{r}_{j'}|\tilde{r}_j > \tilde{r}_{j'}] > C$$

where expectations are taken over all minipatches $F$ of size $m$ in $\mathcal{S}$ such that $j$ and $j'$ are sampled together.

This final assumption ensures that correctly ordered features are separated by a larger margin than incorrectly ordered ones. Specifically, when a more important feature is ranked above a less important one, their expected rank difference exceeds the expected difference when incorrectly ordered by at least some small positive constant $C$. This property provides stability in our estimates where ranking errors have less impact than correct orderings, allowing us to recover true feature ordering through ensembling. Together, these assumptions enable theoretical guarantees for our algorithms.

**Theorem 5.** *Under Assumptions 1-4, if the number of minipatches satisfies*

$$B_{RAMP} = \mathcal{O}\left(\frac{M^3}{m}\ln\left(\frac{kM}{\delta}\right)\right),$$

*then with probability at least $1 - \delta$, RAMP will correctly rank all top-k features: $\hat{r}_{\tau_j} = r_{\tau_j}$ for $j \in \{1, \ldots, k\}$.*

We defer the proof to Appendix A.2. Building on these results, we now show that RAMPART achieves stronger performance guarantees while requiring the same order of computational complexity.

**Theorem 6.** *Suppose $T \geq 3$. Then under assumptions 1-4, there exist choices of $\{B_t\}_{t=1}^T$ such that RAMPART correctly identifies the top-k features with probability at least $1 - \delta/2 - \delta/T$ using the same order of total minipatches as RAMP: $\sum_{t=1}^T B_t \sim B_{RAMP}$.*

We defer the proof to Appendix A.3. While both algorithms require the same order of total minipatches, RAMPART achieves the correct ranking with higher probability. Furthermore, our assumptions are substantially weaker compared to existing work. Traditional multi-armed bandit approaches require rewards to be independent (Russo, 2020), while recent works on variable selection have imposed much stronger identifiability assumptions demanding each arm's optimal reward distribution be uniformly separated from all other possibilities (Liu and Ročková, 2023). In contrast, our analysis does not assume independence among features nor parametric measurement noise: we only require probabilistic consistency and better-than-chance pairwise ordering, allowing for substantial noise and correlation between features.

## 4 Empirical Studies

In this section, we demonstrate the empirical performance of RAMP and RAMPART through a series of carefully designed experiments and two real-data case studies.

### 4.1 Comparative Simulation Studies

We design a comprehensive simulation framework to evaluate feature ranking methods across numerous settings. We first generate data from multivariate normal distributions $\mathbf{x}_i \in \mathbb{R}^M$ under two covariance structures: identity ($\Sigma = I$) or autoregressive ($\Sigma_{i,j} = \rho^{|i-j|}$ with $\rho = 0.5$). For both covariance structures, we assign non-zero coefficients $\beta_i = \gamma(10 - i + 1)$ to the first ten features ($i = 1, \ldots, 10$), with all others set to zero. The parameter $\gamma \in \{0.03, 0.05, 0.10, 0.20, 0.50\}$, referred to as signal-to-noise ratio (SNR) in our subsequent analyses, controls the separation between coefficient values, with smaller values creating more challenging ranking problems.

We construct four distinct settings: linear regression, nonlinear additive regression, linear classification, and nonlinear additive classification. For regression settings, we generate responses as $y_i = f(\mathbf{x}_i) + \epsilon_i$ with $\epsilon_i \sim \mathcal{N}(0,1)$; for classification, $y_i \sim \text{Ber}(1/(1 + e^{-f(\mathbf{x}_i)}))$. The function $f$ distinguishes linear settings, where $f(\mathbf{x}_i) = \mathbf{x}_i^T \beta$, from nonlinear additive settings, where $f(\mathbf{x}_i) = \sum_{j=1}^M \beta_j g_j(\mathbf{x}_{i,j})$, with $g_j(x) = \cos^{j+1}(x)$ for $j \in \{1, ..., 5\}$ and $g_j(x) = \sin^{j-4}(x)$ for $j \in \{6, ..., 10\}$. All features $X_j$ in the linear settings and $g_j(X_j)$ in the nonlinear settings are standardized to zero mean and unit variance, ensuring that coefficient magnitudes $|\beta_m|$ directly reflect the contribution of the features and function as the ground truth importance measures.

In each scenario, we employ task-appropriate prediction models: OLS and logistic regression for the linear regression and classification settings, respectively, and random forests (100 trees) for the nonlinear regression and classification tasks. Additionally, we employ neural networks across all settings, configured as regressors for regression tasks and classifiers (with final sigmoidal activation) for classification tasks. All neural networks have a consistent two-layer architecture with $M$ hidden units and ReLU activation trained to convergence.

As comparison methods, we first include a baseline (model-specific) feature importance (FI) method for each prediction model: absolute coefficients for OLS and logistic regression, Mean Decrease in Impurity (MDI) for random forests, and Integrated Gradients for neural networks (Sundararajan et al., 2017). We also evaluate two popular model-agnostic approaches. First, we apply SHAP with architecture-specific variants (LinearSHAP, TreeSHAP, or GradientSHAP) (Lundberg and Lee, 2017; Lundberg et al., 2020), computing global scores by averaging local attribution scores across observations. Second, we assess permutation importance by measuring the average change in prediction error over 100 random permutations on a held-out set of size $N/2$. For all methods, we obtain feature rankings by sorting importance scores by magnitude.

To ensure a fair and direct comparison, we implement RAMP and RAMPART using the same baseline (model-specific) feature importance method, described previously, for each prediction model: absolute coefficients for OLS and logistic regression, MDI for decision trees, and Integrated Gradients for small neural networks (two-layer neural network with $5m$ hidden units and ReLU activation trained for 5 epochs on each minipatch). This implementation ensures direct comparability, as performance differences stem solely from our algorithmic

framework rather than from variations in feature importance attributions. Both RAMP and RAMPART use minipatches with $n = 125$ observations and $m = 10$ features. For our experiments with $M = 500$ dimensions and $k = 10$ target features, RAMPART requires 6 halving iterations with 2000 minipatches per iteration, while RAMP uses 10000 total minipatches, maintaining comparable computational budgets. Note we omit Goldwasser and Hooker (2025) as their approach requires repeatedly resampling Shapley estimates for top features when statistical tests fail, making it prohibitively expensive in high dimensions.

We evaluate ranking performance using Rank-Biased Overlap (RBO) with $\rho = 0.7$ (Webber et al., 2010), which naturally prioritizes accuracy at higher ranks through geometrically decreasing weights, making it appropriate for our top-$k$ ranking task:

$$\text{RBO}_\rho := (1 - \rho) \cdot \sum_{s=1}^{k} \rho^{s-1} \frac{|\{\hat{\tau}_i\}_{i=1}^{s} \cap \{\tau_i\}_{i=1}^{s}|}{s}$$

For each experimental setting, we conduct 100 simulations with random seeds fixed across all methods, averaging the resulting RBO scores and reporting standard error bars in our plots. Figure 1 demonstrates that RAMPART consistently yields the most accurate feature rankings across both regression and classification tasks, highlighting the benefits of adaptive resource allocation. RAMPART's advantage becomes particularly pronounced at higher signal-to-noise ratios (SNR $\geq 0.1$) and under autoregressive covariance, showcasing robust performance even with correlated features. While performance naturally decreases in nonlinear additive settings for all methods, RAMPART maintains its relative advantage. These results demonstrate that RAMPART's adaptive allocation strategy provides substantial practical benefits across diverse modeling settings. Additional results for higher dimensions ($M = 1000$ and $M = 2000$) with fixed sample size ($n = 250$) are provided in Appendix B, further showcasing RAMPART's robustness in increasingly challenging high-dimensional settings.

To further demonstrate how accurate feature rankings translate to improved predictive performance, we also conducted ablation studies using the same simulation setup as in our identity covariance classification experiments from Figure 1. Here, using the same predictive models and configurations as before, we selected two representative signal-to-noise ratios (SNR = 0.06 and SNR = 0.5), split the data into a 70/30 train-test split, and assessed the model's test prediction performance as the top-ranked features are progressively added in the model in order of their estimated importance rankings. In Figure 2, we illustrate how classification error decreases as the top-ranked features are progressively added as predictors in the model. This comparison reveals the practical impact of feature importance ranking accuracy on prediction performance, with RAMPART's superior rankings consistently yielding the lowest classification errors from models trained using only the top-ranked features. Additional simulation results for ablation studies with higher dimensions can be found in Appendix B, which further verify RAMPART's robustness in increasingly challenging high-dimensional settings.

## 4.2 Cancer Genomics Case Studies

We finally demonstrate RAMPART's effectiveness through two high-dimensional cancer genomics case studies: (i) a regression problem to predict the efficacy or response of a cancer drug on different cancer cell lines, and (ii) a classification problem to predict breast cancer subtypes based upon gene expression profiles. Across both settings, our goal is not only to build accurate predictive models, but to rank the genes that most strongly drive these outcomes, thereby providing interpretable insight into underlying biology and suggesting hypotheses for future therapeutic development.

### 4.2.1 Case Study I: Drug Response Prediction

For our first case study, we predict response to PD-0325901 (an MEK inhibitor) across $N = 259$ human cancer cell lines using their RNASeq gene expression profiles ($M = 1104$ genes) from the Cancer Cell Line Encyclopedia (CCLE) (Barretina et al., 2012) (see Appendix C.1 for details). As before, we compare multiple feature ranking methods to identify the top 10 response-driving genes. Our baseline approach uses a random forest regressor (200 trees) with Mean Decrease in Impurity (MDI). We also compute TreeSHAP values on the same random forest model and evaluate permutation importance by measuring prediction error changes

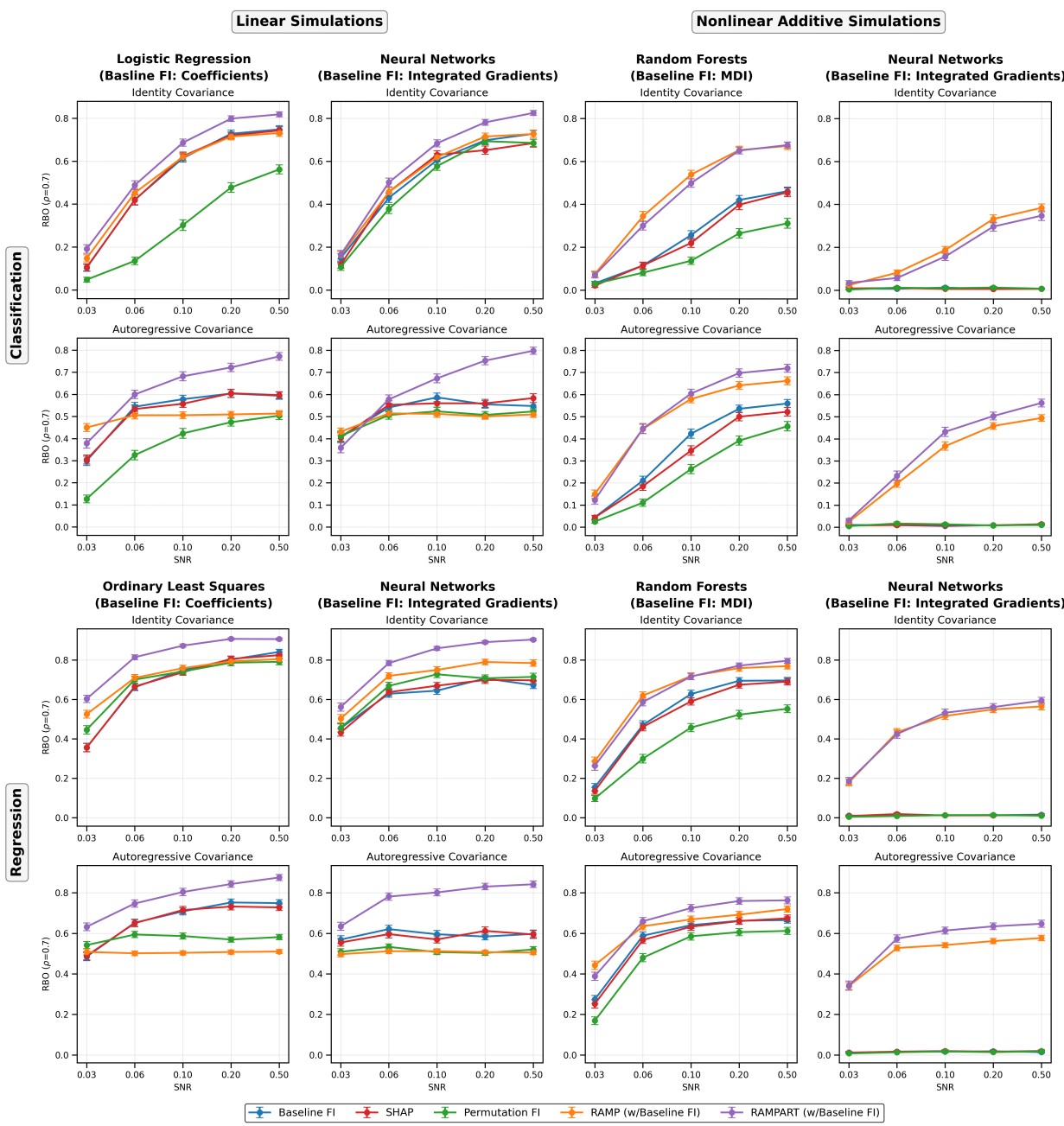

Figure 1: Feature importance ranking accuracy for classification (top) and regression (bottom) ($M = 500$)

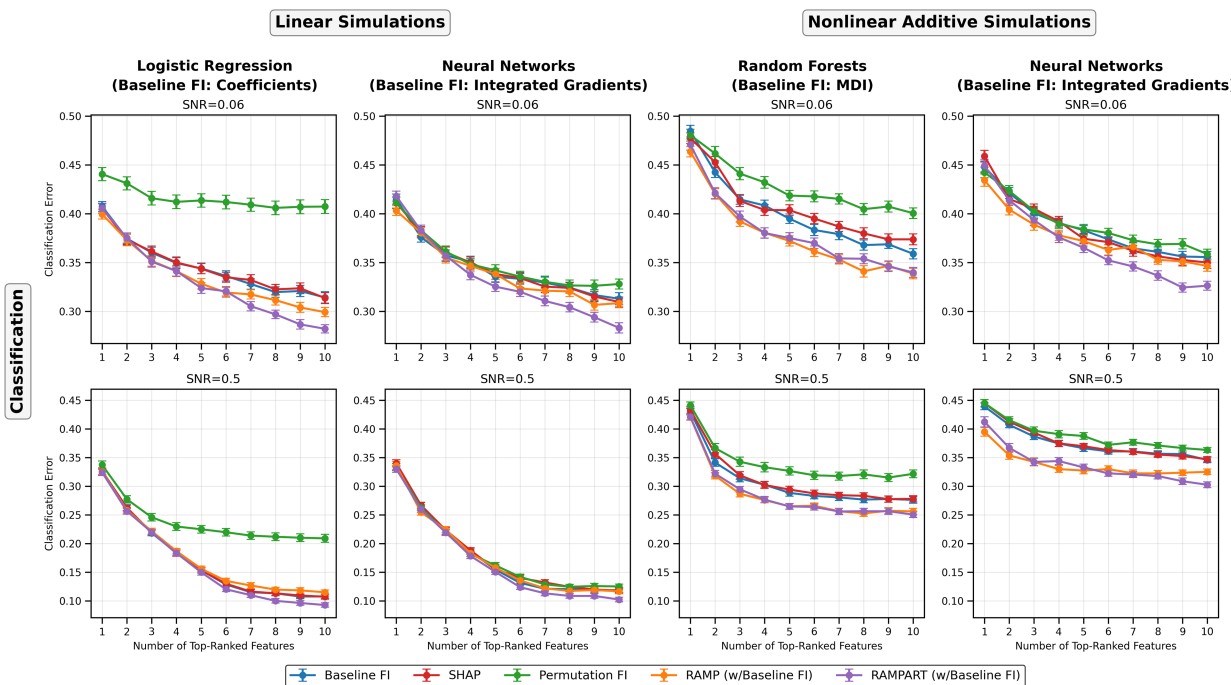

Figure 2: Classification error vs. number of top-ranked features used as predictors in ablation simulation with identity covariance ($M = 500$)

over 100 random permutations on a 50/50 train-test split. For RAMP and RAMPART, we use MDI with regression trees as the minipatch ranking procedure $\mathcal{M}$, with minipatch parameters $m = 10$ and $n = 100$. RAMPART uses 4000 minipatches per iteration, while RAMP used 20000 total minipatches. Table 4 presents the resulting gene rankings from all methods.

Since real-world genomic studies lack ground-truth feature importance rankings, we validate our top-10 gene findings using established biological knowledge. This biological validation approach follows established precedent in the feature importance literature (e.g., see Lundberg et al., 2020; Covert et al., 2020) where feature importance on genomic data are validated through connections to prior medical research. Along these lines, there are several key biological findings from this case study. First, all 10 genes identified by RAMPART have been previously implicated in biological pathways involving PD-0325901 and related cancers (Table 1). In particular, the top-ranked gene from RAMPART, *TOR4A*, is a known oncogene for glioma (Wang et al., 2022), a cancer for which PD-0325901 is currently being tested in clinical trials (Vinitsky et al., 2022). Second, the main pathway affected by PD-0325901 is the MEK/ERK signaling pathway, which regulates cell proliferation, differentiation, and survival. Several of the genes identified by RAMPART, including *ETV4*, *SPRY2*, and *WNT5A*, are key factors in the MEK/ERK pathway (Oh et al., 2012; Milillo et al., 2015; Hasan et al., 2021).

Finally, we perform a gene ontology (GO) enrichment analysis (Ashburner et al., 2000; Aleksander et al., 2023), a standard bioinformatics approach for identifying the biological processes that are enriched (or over-represented) in a set of genes, to further validate the biological relevance of the top-10 genes identified by RAMPART. According to the GO enrichment analysis, the biological processes, "regulation of transmembrane receptor protein serine/threonine kinase signaling pathway" and "regulation of cellular response to growth factor stimulus," are significantly enriched (FDR $p < 0.05$) in the top-10 genes from RAMPART. Notably, the serine/threonine kinase signaling pathway plays an essential role in the activation of MEK (Zheng and Guan, 1994). In contrast, no GO biological processes are significantly enriched in the top-10 genes from any of the other competing methods. Additionally, of the four genes (i.e., *SPRY2*, *FERMT1*, *WNT5A*, and *NRROS*) linked to the identified GO biological processes, two are uniquely identified by RAMPART and have not been ranked in the top 10 by any other method. This GO enrichment analysis not only supports

| Rank | Gene | Connection to PD-0325901 |
|------|------|--------------------------|
| 1 | *TOR4A* | Torsin family gene and oncogene for glioma and other cancers (Wang et al., 2022) |
| 2 | *ETV4* | Well-known ETS transcription factor regulated by MEK/ERK pathway (Oh et al., 2012) |
| 3 | *SPRY2* | Key inhibitor of the MEK/ERK pathway (Zheng and Guan, 1994; Milillo et al., 2015) |
| 4 | *GJB1* | Gap junction gene associated with increased tumor progression for various cancers (Aasen et al., 2016) |
| 5 | *PYCARD* | Encodes key adaptor protein in inflammatory and apoptotic signaling pathways, playing dual roles in multiple cancers (Protti and De Monte, 2020) |
| 6 | *WNT5A* | Wnt signaling pathway gene which can enhance MEK/ERK pathway (Zheng and Guan, 1994; Hasan et al., 2021) |
| 7 | *FERMT1* | Regulates Ser/Thr kinase signaling pathway, activating MEK (Zheng and Guan, 1994) |
| 8 | *NRROS* | Regulates Ser/Thr kinase signaling pathway, activating MEK (Zheng and Guan, 1994) |
| 9 | *LYZ* | Encodes lysozyme; shown to exhibit aberrant expression in tumor cells (Gu et al., 2023) |
| 10 | *NPAS2* | Regulator of circadian rhythms and tumor suppressor involved in DNA damage response (Hoffman et al., 2008) |

Table 1: Top-10 genes from RAMPART for drug response prediction and their connections to PD-0325901. Genes highlighted in blue were identified by the GO enrichment analysis to be involved in the "regulation of transmembrane receptor protein serine/threonine (Ser/Thr) kinase signaling pathway" and "regulation of cellular response to growth factor stimulus."

the biological relevance of RAMPART's identified genes, but also highlights the advantage of RAMPART over existing methods for feature importance ranking and high-impact scientific discovery.

### 4.2.2 Case Study II: Breast Cancer Subtype Classification

We next consider a high-dimensional multi-class classification problem based on The Cancer Genome Atlas (TCGA) Breast Cancer (BRCA) cohort. Using RNA-seq data from the TCGA–BRCA study (The Cancer Genome Atlas Network, 2012) and PAM50 subtype labels (Parker et al., 2009), we obtain a gene-expression matrix with $N = 758$ primary tumors and $M = 5000$ genes after preprocessing (see Appendix C.2 for details). Each sample is assigned one of five intrinsic subtype labels (Luminal A, Luminal B, HER2-enriched, Basal-like, Normal-like), yielding a five-class classification outcome. As in Section 4.2.1, we compare multiple feature-ranking methods and focus on the top 10 genes from each method as subtype-discriminative markers. All hyperparameters are identical to Case Study 4.2.1, except that we replace decision-tree/random-forest regressors with their classification counterparts. Table 5 presents the resulting gene rankings from all methods.

Notably, the 10 RAMPART-selected genes in Table 2 have each been implicated and extensively studied in the previous breast cancer literature. Genes including *ESR1*, *FOXA1*, *GATA3*, and *MLPH* mark luminal, estrogen receptor–driven disease and are tightly linked to endocrine response and prognosis (Brett et al., 2021; Toy et al., 2013; Fu et al., 2016; Hurtado et al., 2011; Takaku et al., 2015; Kouros-Mehr et al., 2006; Thakkar et al., 2015). *FOXC1* captures the basal-like transcriptional programme and loss of luminal identity (Han et al., 2017; Yu-Rice et al., 2016). On the other hand, *TPX2*, *ASPM*, *CDK1*, *FOXM1*, and *UBE2C* function as core regulators of mitotic entry, spindle assembly, and chromosome segregation whose overexpression marks highly proliferative, poor-prognosis tumors (Marugán et al., 2024; Tang et al., 2019; Enserink and Chymkowitch, 2022; Bergamaschi et al., 2014; Hu et al., 2025).

We also performed GO Biological Process enrichment on the top-10 genes from each method and visualized the processes significantly enriched at FDR $p \leq 0.05$ (Table 6). RAMPART's genes are enriched for 16 such biological processes, including transcriptional regulation, reproductive and gland-related development, and cell-cycle and proliferation-associated programs, which are well-known components in cancer biology (Hanahan and Weinberg, 2011). In contrast, RAMP, SHAP, and permutation importance identify a more limited set of 5, 10, and 6 enriched GO terms, respectively.

| Rank | Gene | Role in Breast Cancer Biology |
|------|------|-------------------------------|
| 1 | ESR1 | Encodes estrogen receptor-$\alpha$; ESR1 mutations and dysregulation drive endocrine resistance in ER$^+$ disease (Brett et al., 2021; Toy et al., 2013) |
| 2 | FOXC1 | Basal-like transcription factor opposing GATA3/ER$\alpha$; linked to ER$\alpha$ loss, endocrine resistance, and aggressive basal-like tumors (Yu-Rice et al., 2016) |
| 3 | TPX2 | Mitotic spindle assembly factor overexpressed in CIN-high tumors; activates YAP signaling and sensitizes cells to the SRC inhibitor dasatinib (Marugán et al., 2024) |
| 4 | FOXA1 | Pioneer transcription factor opening chromatin for ER–DNA binding; reprograms ER-driven transcription and promotes endocrine resistance (Fu et al., 2016) |
| 5 | ASPM | Spindle pole and centrosome protein; overexpression drives proliferation and marks high-grade, poor-prognosis breast cancer (Tang et al., 2019) |
| 6 | MLPH | Component of an estrogen-responsive luminal signature; high MLPH marks ER$^+$ tumors with good prognosis (Thakkar et al., 2015) |
| 7 | CDK1 | Core cyclin-dependent kinase controlling $G_2$/M transition and mitotic entry; hyperactivation marks highly proliferative tumors (Enserink and Chymkowitch, 2022) |
| 8 | FOXM1 | Oncogenic transcription factor driving $G_2$/M cell-cycle genes; promotes proliferation, invasion, and endocrine resistance in ER$^+$ disease (Bergamaschi et al., 2014) |
| 9 | GATA3 | Master regulator of mammary luminal differentiation; recurrently mutated in luminal tumors and essential for luminal identity (Kouros-Mehr et al., 2006; Takaku et al., 2015) |
| 10 | UBE2C | Ubiquitin-conjugating enzyme activating the APC/C complex; overexpressed in highly proliferative tumors and associated with poor prognosis (Hu et al., 2025) |

Table 2: Top-10 genes from RAMPART for breast cancer subtype prediction and their functional roles.

| Method | Case Study I | Case Study II |
|--------|-------------|---------------|
| RAMPART | 21.83 | 36.97 |
| RAMP | 14.15 | 20.65 |
| Baseline | 12.48 | 3.62 |
| TreeSHAP | 13.79 | 58.46 |
| Permutation | 489.86 | 3637.54 |

Table 3: Wall-clock time (seconds) for computing feature rankings in Case Study I (drug response, $N = 259$, $M = 1104$ genes) and Case Study II (breast cancer subtypes, $N = 758$, $M = 5000$ genes). All methods use the same tree-ensemble hyperparameters across both datasets; times exclude data loading and basic preprocessing. Experiments were run on a 16-inch MacBook Pro (Apple M3 Max, 48 GB RAM, macOS 14.3).

## 4.3 Runtime and Scalability

**Practical runtime.** Table 3 shows that RAMPART is computationally lightweight in the regimes we consider. In the smaller CCLE drug-response case study ($M = 1104$), all non-permutation methods finish within 25 seconds: RAMPART takes 21.8 seconds, compared with 12–14 seconds for Baseline RF and TreeSHAP. Thus, on this moderate-dimensional problem, the extra stability we gain from recursive trimming comes at only a modest constant-factor overhead. When we move to the higher-dimensional TCGA breast cancer subtype task ($M = 5000$) keeping all hyperparameters fixed, RAMP and RAMPART remain well under a minute (20.7 and 37.0 seconds), and now RAMPART is actually *faster* than TreeSHAP (58.5 seconds) while providing a more robust top-$k$ ranking. In both case studies, permutation feature importance is one to two orders of magnitude slower (from $\sim 8$ minutes up to more than an hour), reflecting the need to re-evaluate the trained forest on many permuted versions of each feature and thereby perform work that scales essentially linearly with the feature dimension and the number of permutations.

**Scaling with dimensionality.** The scalability of RAMPART follows from how we allocate compute across trimming rounds. Each round fits the same number of minipatch forests on patches of fixed size and on a fixed subsample of observations, so its cost is approximately a constant multiple of the cost of fitting a single random forest on a small feature subset. The only quantity that grows is the number of rounds $L$ needed to shrink from $M$ candidates down to $k$ finalists: with a halving schedule, $L \approx \lceil \log_2(M/k) \rceil$. In contrast, RAMP runs only a single such round (with more minipatches), so RAMPART incurs an additional logarithmic factor in runtime relative to RAMP while sharing the same per-round cost. Thus, the dominant term in the runtime of RAMPART is a constant times $\log(M/k)$. In our experiments, increasing $M$ by a factor of $\approx 4.5$ (from 1104 to 5000) increases RAMPART's runtime by only a factor of $\approx 1.7$, whereas TreeSHAP slows down by more than $4\times$ and permutation by more than $7\times$. Taken together, these results illustrate the core advantage of our design: RAMPART attains a statistically stronger and more interpretable top-$k$ ranking at the cost of only a small, logarithmically growing multiple of fitting a single forest, while remaining scalable to increasingly higher-dimensional settings.

## 5 Discussion

In this paper, we introduced RAMPART, a novel framework that achieves accurate feature ranking in high-dimensional settings by combining minipatch ensembles with recursive trimming. Instead of developing a new feature importance method, we adopt a complementary, model-agnostic view that can leverage any existing feature importance procedure as a black-box primitive. RAMPART wraps these existing feature importance procedures in a ranking algorithm that uses inexpensive subsampled fits to screen many features and concentrate computation on the candidates that plausibly belong in the top-$k$. As demonstrated through our genomics case studies, precise feature ranking can drive scientific discovery by identifying key drivers in complex biological processes and guiding future research. However, while our empirical and theoretical results are encouraging, they also highlight several challenges and opportunities. Given that RAMPART is a wrapper method, it may inherit modeling assumptions and potential biases from the underlying attribution procedure, and our current guarantees are derived under stylized separation and consistency conditions that may be difficult to verify in practice. These considerations motivate several directions for future work.

First, adapting minipatch sizes and sampling schemes as the candidate feature pool shrinks could enable more precise comparisons while maintaining computational efficiency, and our minipatch ensemble framework offers a flexible foundation for exploring alternative adaptive sampling methods to further enhance top-$k$ ranking performance. Second, extending the ideas developed here beyond tabular settings to other data modalities, such as graphs, images, and time series, via domain-appropriate base explainers would broaden the impact of this paradigm. Finally, refining our assumptions and sharpening theoretical bounds in regimes that mirror high-stakes applications would further strengthen the reliability of RAMPART for scientific and decision-making use.

### Acknowledgments

GA and YC acknowledge support from NSF DMS-2516872.

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

## A Proofs

### A.1 Auxiliary Lemmas

**Lemma 7.** *Let $\mu_j = \mathbf{E}[\tilde{r}_j]$ be the expected rank of feature $j$ under the ranking procedure $\mathcal{M}$, where the expectation is taken over minipatches that sample feature $j$. Next, define*

$$\Delta := \min_{j,j' \in [M], r_j < r_{j'}} \mu_{j'} - \mu_j$$

*to be the smallest difference between the expected rank estimates across all features in $[M]$. Then under assumptions 2, 3 and 4, we have*

$$\Delta > (2p - 1) \cdot \left( \frac{m-1}{M-1} \right).$$

*Proof.* First, observe that given the above assumptions, we necessarily have $\mu_j < \mu_{j'}$ whenever $r_j < r_{j'}$ as we have

$$\begin{aligned}
\mu_j &= \mathbf{E}[\tilde{r}_j | j \in F] \\
&= \mathbf{E}[\tilde{r}_j | j \in F, j' \notin F] \cdot \mathbf{P}(j' \notin F \mid j \in F) + \mathbf{E}[\tilde{r}_j | j, j' \in F] \cdot \mathbf{P}(j' \in F \mid j \in F) \\
&< \mathbf{E}[\tilde{r}_{j'} | j' \in F, j \notin F] \cdot \mathbf{P}(j \notin F \mid j' \in F) + \mathbf{E}[\tilde{r}_{j'} | j, j' \in F] \cdot \mathbf{P}(j \in F \mid j' \in F) \\
&= \mathbf{E}[\tilde{r}_{j'} | j' \in F] \\
&= \mu_{j'}
\end{aligned}$$

where the inequality follows from knowing that $\mathbf{E}[\tilde{r}_{j'} | j' \in F, j \notin F] > \mathbf{E}[\tilde{r}_j | j \in F, j' \notin F]$ by assumption 3, that $\mathbf{P}(j' \notin F \mid j \in F) = \mathbf{P}(j \notin F \mid j' \in F)$, and that $\mathbf{E}[\tilde{r}_{j'} | j, j' \in F] > \mathbf{E}[\tilde{r}_j | j, j' \in F]$ by assumptions 2 and 4 since

$$\begin{aligned}
&\mathbf{P}(\tilde{r}_{j'} > \tilde{r}_j | j, j' \in F) \cdot \left( \mathbf{E}[\tilde{r}_{j'} | \tilde{r}_{j'} > \tilde{r}_j] - \mathbf{E}[\tilde{r}_j | \tilde{r}_{j'} > \tilde{r}_j] \right) \\
&> \mathbf{P}(\tilde{r}_j > \tilde{r}_{j'} | j, j' \in F) \cdot \left( C + \mathbf{E}[\tilde{r}_j | \tilde{r}_j > \tilde{r}_{j'}] - \mathbf{E}[\tilde{r}_{j'} | \tilde{r}_j > \tilde{r}_{j'}] \right)
\end{aligned}$$

and rearranging gives the desired result. It follows that for any $j, j' \in [M]$ with $r_j < r_{j'}$

$$\begin{aligned}
\mu_{j'} - \mu_j &= \mathbf{E}[\tilde{r}_{j'} | j, j' \in F] \cdot \mathbf{P}(j \in F \mid j' \in F) + \mathbf{E}[\tilde{r}_{j'} | j' \in F, j \notin F] \cdot \mathbf{P}(j \notin F \mid j' \in F) \\
&\quad - \mathbf{E}[\tilde{r}_j | j, j' \in F] \cdot \mathbf{P}(j' \in F \mid j \in F) - \mathbf{E}[\tilde{r}_j | j \in F, j' \notin F) \cdot \mathbf{P}(j' \notin F \mid j \in F) \\
&> \mathbf{E}[\tilde{r}_{j'} - \tilde{r}_j | j, j' \in F] \cdot \mathbf{P}(j \in F \mid j' \in F) \\
&= \left( \mathbf{E}[\tilde{r}_{j'} - \tilde{r}_j | \tilde{r}_{j'} > \tilde{r}_j] \cdot \mathbf{P}(\tilde{r}_{j'} > \tilde{r}_j) + \mathbf{E}[\tilde{r}_{j'} - \tilde{r}_j | \tilde{r}_j > \tilde{r}_{j'}] \cdot \mathbf{P}(\tilde{r}_j > \tilde{r}_{j'}) \right) \cdot \mathbf{P}(j \in F \mid j' \in F) \\
&> \left( \mathbf{E}[\tilde{r}_{j'} - \tilde{r}_j | \tilde{r}_{j'} > \tilde{r}_j] \cdot p + (C - \mathbf{E}[\tilde{r}_{j'} - \tilde{r}_j | \tilde{r}_{j'} > \tilde{r}_j]) \cdot (1 - p) \right) \cdot \mathbf{P}(j \in F \mid j' \in F) \\
&= \left( \mathbf{E}[\tilde{r}_{j'} - \tilde{r}_j | \tilde{r}_{j'} > \tilde{r}_j] \cdot (2p - 1) + C(1 - p) \right) \cdot \mathbf{P}(j \in F \mid j' \in F) \\
&> (2p - 1) \cdot \mathbf{P}(j \in F \mid j' \in F) \\
&> (2p - 1) \cdot \frac{(m-1)}{(M-1)}
\end{aligned}$$

where the first equality holds by the law of total conditional expectation, the first inequality holds by assumption 3, the second inequality holds by assumption 4, and the second-to-last inequality holds since the difference between estimated ranks is at least 1 and that $C > 0$. $\square$

**Lemma 8.** *For iteration $t$ of algorithm 2, define the minimum separation gap*

$$\Delta_t := \min_{j,j' \in \mathcal{C}_t, r_j < r_{j'}} \mu_j^t - \mu_{j'}^t$$

*where $\mu_j^t$ is the expected rank estimate of feature $j$ with respect to the feature pool $\mathcal{C}_t$. Under Assumptions 2-4, we have*

$$\Delta_t > 2^{t-1} \cdot (2p - 1) \cdot \left( \frac{m-1}{M-1} \right)$$

*for any iteration $t$.*

Intuitively, Lemma 8 shows that trimming increases the expected gap between adjacent ranks geometrically across rounds, enabling finer discrimination among near-tied features with the same overall sampling budget.

*Proof.* At any iteration $t$, we know from lemma 7 that

$$\Delta_{t+1} > (2p-1) \cdot \frac{m-1}{|C_{t+1}|-1} = (2p-1) \cdot \frac{m-1}{M/2^t - 1} \geq 2^t \cdot (2p-1) \cdot \frac{m-1}{M-1}.$$

$\square$

### A.2 Proof of Theorem 5

*Proof.* Let $\mathcal{T} = \{\tau_1, \ldots, \tau_k\}$ denote the set of top-$k$ features. In addition, for any feature $j \in [M]$ let $S_j$ denote the set of minipatches where feature $j$ is included. Next, for any given feature $j$ we order the minipatches in $S_j$ arbitrarily as $\kappa_1, \ldots, \kappa_{n_j}$ where $n_j = |S_j|$ and define the martingale difference sequence indexed by

$$X_s = \tilde{r}_j^{\kappa_s} - \mathbf{E}[\tilde{r}_j^{\kappa_s}|\mathcal{F}_{s-1}]$$

where $\mathcal{F}_{s-1} = \sigma(\kappa_1, \ldots, \kappa_{s-1})$. We see that $X_s$ is $\mathcal{F}_s$-measurable and

$$\mathbf{E}[X_s|\mathcal{F}_{s-1}] = \mathbf{E}[\tilde{r}_j^{\kappa_s} - \mathbf{E}[\tilde{r}_j^{\kappa_s}|\mathcal{F}_{s-1}]|\mathcal{F}_{s-1}] = 0.$$

We also trivially see that $|X_s| \leq m-1$. Then, by Azuma-Hoeffding's inequality we can write

$$\mathbf{P}(\bar{r}_j - \mu_j \geq \epsilon) = \mathbf{P}(M_{n_j} \geq n_j \epsilon) \leq \exp\left(-\frac{n_j^2 \epsilon^2}{2n_j(m-1)^2}\right) = \exp\left(-\frac{n_j \epsilon^2}{2(m-1)^2}\right)$$

where the first equality holds by observing that

$$M_{n_j} := \sum_{s=1}^{n_j} X_s = \sum_{s=1}^{n_j} \tilde{r}_j^{\kappa_s} - \mathbf{E}[\tilde{r}_j^{\kappa_s}|\mathcal{F}_{s-1}] = n_j(\bar{r}_j - \mu_j).$$

Next, since $n_j \sim \text{Binomial}(B, q)$ where $q = m/M$, we have by the Chernoff bound that

$$\mathbf{P}(n_j \leq \frac{Bq}{2}) \leq \exp(-\frac{Bq}{8}).$$

Then, by the law of total probability we can write

$$\mathbf{P}(\bar{r}_j - \mu_j \geq \epsilon) = \mathbf{P}\left(\bar{r}_j - \mu_j \geq \epsilon, n_j \leq \frac{Bq}{2}\right) + \mathbf{P}\left(\bar{r}_j - \mu_j \geq \epsilon, n_j \geq \frac{Bq}{2}\right)$$

$$\leq \exp\left(-\frac{Bq}{8}\right) + \mathbf{P}\left(\bar{r}_i - \mu_i \geq \epsilon, n_i \geq \frac{Bq}{2}\right)$$

$$\leq \exp\left(-\frac{Bq}{8}\right) + \exp\left(-\frac{Bq\epsilon^2}{4(m-1)^2}\right).$$

As before, we define

$$\Delta := \min_{j,j' \in [M], r_j < r_{j'}} \mu_{j'} - \mu_j$$

and set $\epsilon = \Delta/2$. By assumption 1 this quantity is non-zero and well-defined. Then we can write for some top-$k$ feature $j$ and other feature $j'$ where $r_j < r_{j'}$ that

$$\mathbf{P}(\bar{r}_j \geq \bar{r}_{j'}) \leq \mathbf{P}(\bar{r}_j \geq \mu_j + \epsilon) + \mathbf{P}(\bar{r}_{j'} \leq \mu_{j'} - \epsilon).$$

We see that the probability of any true top-$k$ feature not being ranked correctly is upper bounded by

$$\mathbf{P}_{\text{err}} \leq \mathbf{P}\left(\bigcup_{j \in \mathcal{T}} \bigcup_{j':r_{j'}>r_j} \{\bar{r}_j \geq \bar{r}_{j'}\}\right).$$

By the union bound, we see we want to find $B$ such that

$$\mathbf{P}_{\text{err}} \leq \sum_{j \in \mathcal{T}} \sum_{j':r_{j'}>r_j} \mathbf{P}(\bar{r}_j \geq \bar{r}_{j'}) \leq 2kM \cdot \left[\exp\left(-\frac{Bq}{8}\right) + \exp\left(-\frac{Bq\epsilon^2}{4(m-1)^2}\right)\right] \leq \delta.$$

One way to choose $B$ is such that each of the two exponentials $\leq \delta/4kM$. Solving for these two conditions separately gives

$$B \geq \max\left\{\frac{8}{q}\ln\left(\frac{4kM}{\delta}\right), \frac{4(m-1)^2}{q\epsilon^2}\ln\left(\frac{4kM}{\delta}\right)\right\}$$
$$= \frac{4(m-1)^2}{q\epsilon^2}\ln\left(\frac{4kM}{\delta}\right)$$
$$= \frac{16M(M-1)^2}{(2p-1)^2 m}\ln\left(\frac{4kM}{\delta}\right)$$

where $\epsilon = \Delta/2$ and $\Delta \geq (2p-1)\left(\frac{m-1}{M-1}\right)$ by lemma 7. Choosing $C = \frac{16}{(2p-1)^2}$ yields the result. $\qquad\square$

## A.3 Proof of Theorem 6

*Proof.* Let $N_t = M/2^{t-1}$ denote the feature pool size at iteration $t$, $q_t = m/N_t$, $\epsilon_t = \delta_t/2$, and $\mathcal{D}_t$ be the features ranked in the bottom half at iteration $t$ to be discarded. We trivially see that $r_{j'} > r_j$ for any $j' \in \mathcal{D}_t$ and $j \in \mathcal{T}$. Furthermore, any top-$k$ feature $j$ at iteration $t \leq T-1$ will survive onto the next round by being ranked in the upper-half of the feature pool. Then defining $E_{j,t}$ as the event that feature $j$ is incorrectly eliminated at iteration $t$, we can write for any $t \leq T-1$ that

$$\mathbf{P}(E_{j,t}) \leq \mathbf{P}\left(\bigcup_{j' \in \mathcal{D}_t}\{\bar{r}_j^t \geq \bar{r}_{j'}^t\}\right)$$
$$\leq \sum_{j' \in \mathcal{D}_t}\mathbf{P}(\bar{r}_j^t \geq \bar{r}_{j'}^t)$$
$$\leq \frac{N_t}{2}\left(\mathbf{P}(\bar{r}_j^t \geq \mu_j^t + \epsilon_t) + \mathbf{P}(\bar{r}_{j'}^t \leq \mu_{j'}^t - \epsilon_t)\right)$$
$$= N_t\left[\exp\left(\frac{-B_t q_t}{8}\right) + \exp\left(-\frac{B_t q_t \epsilon_t^2}{4(m-1)^2}\right)\right]$$

where the second inequality holds by the subadditivity of measure, and the last two (in)equalities hold by a similar argument found in appendix A.2. For each iteration $t \leq T-1$, we see that the probability of error is upper bounded by

$$\mathbf{P}_{\text{err}}^t \leq \sum_{j \in \mathcal{T}}\mathbf{P}(E_{j,t}).$$

Whereas for the last iteration $T$, we need to correctly rank all top-$k$ features. Similarly following appendix A.2, we have

$$\mathbf{P}_{\text{err}}^T \leq \sum_{j \in \mathcal{T}}\mathbf{P}(E_{j,T}) \leq \sum_{j \in \mathcal{T}}\sum_{j':r_{j'}>r_j}\mathbf{P}(\bar{r}_j \geq \bar{r}_{j'}).$$

Next, we want to control each term $\mathbf{P}_{\text{err}}^t \leq \delta/2T$ and $\mathbf{P}_{\text{err}}^T \leq \delta/T$ such that the total probability of error is bounded within $\mathbf{P}_{\text{err}} \leq \delta/2 + \delta/T$. To achieve this, and again following a similar argument as outlined in appendix A.2, we can choose

$$B_t \geq \frac{4(m-1)^2}{q_t \epsilon_t^2}\ln\left(\frac{4kTN_t}{\delta}\right)$$

for $t \leq T - 1$ and

$$B_T \geq \frac{4(m-1)^2}{q_t \epsilon_t^2} \ln \left( \frac{4kTN_T}{\delta} \right).$$

Set $\Delta_1 = (2p - 1) \frac{m-1}{M-1}$ Then following lemma 8, we can write $B_{\text{RAMPART}}$ as

$$
\begin{aligned}
B_{\text{RAMPART}} &\leq \sum_{t=1}^{T} B_t = \frac{16(m-1)^2}{m} \sum_{t=1}^{T} \frac{N_t}{\Delta_t^2} \ln \left( \frac{4kTN_t}{\delta} \right) \\
&= \frac{16(m-1)^2}{m\Delta_1^2} \sum_{t=1}^{T} \frac{N_t}{(2^{t-1})^2} \ln \left( \frac{4kTN_t}{\delta} \right) \\
&= \frac{16(m-1)^2}{m\Delta_1^2} \sum_{t=1}^{T} \frac{M}{2^{2(t-1)+t-1}} \left[ \ln \left( \frac{4kTM}{\delta} \right) - (t-1)\ln 2 \right] \\
&= \frac{16(m-1)^2}{m\Delta_1^2} \cdot \left[ M \ln \left( \frac{4kTM}{\delta} \right) \sum_{t=1}^{T} \frac{1}{2^{3t-3}} - M \ln 2 \sum_{t=1}^{T} \frac{(t-1)}{2^{3t-3}} \right] \\
&\leq \frac{16M(m-1)^2}{m\Delta_1^2} \cdot \left[ \frac{6}{5} \ln \left( \frac{4kTM}{\delta} \right) - \frac{1}{10} \ln 2 \right] \\
&\leq \frac{16M(m-1)^2}{m\Delta_1^2} \cdot \left[ \frac{6}{5} \ln \left( \frac{4k(\log_2(M) - \log_2(k) + 1)M}{\delta} \right) \right] \\
&\sim \frac{16M(m-1)^2}{m\Delta_1^2} \cdot \left[ \frac{6}{5} \ln \left( \frac{4kM}{\delta} \right) \right]
\end{aligned}
$$

where in the second-to-last inequality follows from knowing that $T \geq 3$. $\qquad\square$

## B Additional Simulation Results

In this section, we further validate RAMPART's effectiveness and our theoretical analysis by examining how the probability of exactly recovering the true top-$k$ features scales with the total number of minipatches. We then assess robustness in higher dimensions ($M = 1000, 2000$) and analyze the impact of minipatch size on ranking accuracy.

### B.1 Theory Validation

We first verify our theoretical results stated in Theorem 6 by examining how the number of minipatches affects ranking performance of RAMP and RAMPART. We generate synthetic data with $N = 1000$ observations and $M = 160$ features drawn independently from a standard normal distribution $\mathbf{X} \sim \mathcal{N}(0, I)$ with $\mathbf{Y} = 0.22 \cdot \mathbf{X}\beta + \epsilon$, where $\epsilon$ is unit Gaussian noise. The coefficient vector $\beta$ is constructed to have four non-zero features with coefficients 4, 3, 2, 1. Since features are on the same scale, the magnitude of these coefficients directly determines the feature importance ordering, providing a clear ground truth for evaluating ranking performance. We compare RAMP and RAMPART with minipatch parameters $n = 80$ and $m = 20$ (the trimming process also terminates when there are 20 features remaining). To ensure a fair comparison, we allocate a total budget of $B$ minipatches across the five halving iterations for RAMPART, while using $5B$ minipatches for RAMP to match the total computation. We evaluate the empirical success probability averaged over 500 random trials, where success is defined as exactly recovering the ranks of the top four features.

Figure 3 shows the empirical success probability as a function of the total number of minipatches. In general, both methods demonstrate increasing accuracy with more minipatches, with RAMPART consistently achieves superior ranking accuracy compared to RAMP, validating the consistency guarantees of Theorems 5 and 6.

While our theory shows RAMPART and RAMP require the same order of total minipatches, empirically RAMPART achieves better performance, particularly in the regime of 100-1000 minipatches where its adaptive resource allocation proves beneficial. This suggests that RAMPART's strategy of progressively

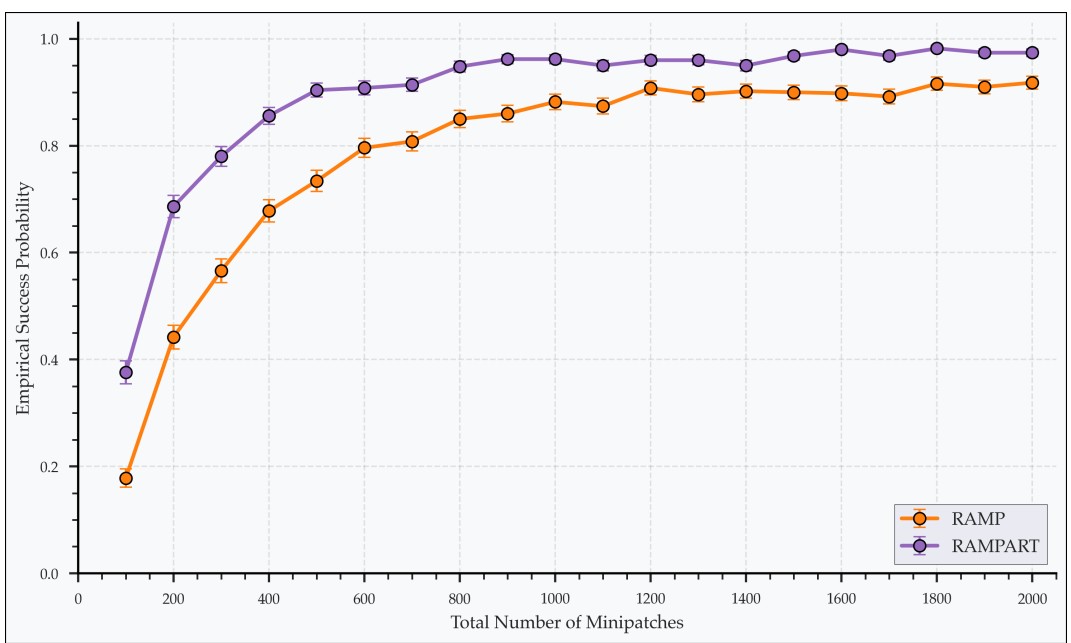

Figure 3: Number of Minipatches vs Ranking Success Probability

focusing computation on promising features provides practical advantages beyond what our theoretical analysis captures. Bridging this gap to obtain sharper theoretical guarantees that better align with empirical performance remains an exciting direction for future work.

## B.2 Higher Dimensions

For $M = 1000$ and 2000, we extend the setup described in Section 4 by adding more nonzero features while preserving the same coefficient structure. We make two implementation adjustments to accommodate higher dimensionality: (1) For baseline FI, SHAP, and feature permutation with random forests, we scale the number of trees (200 trees for $M = 1000$ and 400 trees for $M = 2000$) to ensure adequate feature space coverage; (2) For RAMPART and RAMP, we set $B = 4000$ and $B = 20000$ minipatches respectively, while keeping all other parameters unchanged.

The results for $M = 1000$ (Figures 4 & 5) demonstrate RAMPART's robust performance across both covariance structures, with its advantage becoming particularly pronounced under autoregressive covariance. RAMPART's sequential halving strategy proves especially effective in these higher-dimensional settings, showing remarkable stability compared to $M = 500$ while other methods exhibit noticeable degradation. At extreme dimensionality ($M = 2000$, Figures 6 & 7), RAMPART continues to excel, maintaining strong performance even at high signal-to-noise ratios under autoregressive covariance, while competing methods show significant accuracy drops. These results highlight RAMPART's unique ability to handle both high dimensionality and complex feature interactions through its adaptive resource allocation strategy.

We also extend our ablation studies to higher dimensions using the same methodology as Section 4.1, focusing on the identity covariance setting and examining columns 2 and 5 (SNR = 0.06 and SNR = 0.5) across both classification and regression tasks. As before, we maintain consistent experimental conditions by using the same predictive models, data generation processes, and hyperparameter configurations across all settings. For regression tasks, we measure the mean squared error (MSE) rather than classification error. Figures 8-12 demonstrate that RAMPART maintains its performance advantage across all dimensions, affirming that RAMPART's adaptive resource allocation strategy remains valuable even in higher-dimensional settings where feature ranking becomes more challenging.

### B.3 Impact of Minipatch Size

We also investigate the effect of varying feature subsample size $m$ within the RAMP framework, returning to the $M = 500$ setting with experimental conditions from Section 4. Figures 13 and 14 demonstrate that while performance remains stable across minipatch sizes for linear models under identity covariance, smaller minipatches ($m = 10$) consistently outperform larger ones ($m = 50$) in nonlinear additive and correlated settings. These results suggest that smaller minipatch sizes may better capture local feature interactions while avoiding noise from irrelevant features. While we fix $m = 10$ for RAMPART in this work, exploring adaptive minipatch sizes that scale with feature pool size across halving iterations remains an interesting direction for future research.

## C Additional Case Study Discussions

### C.1 Drug Response Prediction

**Response Variable.** To quantify the PD-0325901 drug response in our case study, we use the area under the dose-response curve (AUC) as the primary outcome of interest. The AUC is a widely-used measure of overall drug sensitivity, defined as the area between the dose-response curve and 0 (Barretina et al., 2012). A higher value indicates that the drug was more effective at killing the cancer cells. We refer to Barretina et al. (2012) for details on how this dose-response data was collected and processed.

**Data Preprocessing of Gene Expression Data.** The raw CCLE data used in this case study can be downloaded from the DepMap Portal (`https://depmap.org/portal/download/`) (version 18Q3). Due to the heavy right-skewed distribution of the RNASeq gene expression values, we log-transformed ($\log(x+1)$) the raw gene expression data. We also restricted our analysis to the 1200 genes with the highest empirical variance across the cell lines and sequentially removed genes which had $> 0.95$ Pearson correlation with another gene in the dataset, resulting in 1104 genes. The processed gene expression data was finally standardized to have zero mean and unit variance.

| Gene | RAMPART | RAMP | Baseline | SHAP | Permutation |
|---|---|---|---|---|---|
| TOR4A | 1 | - | 4 | 2 | 1 |
| ETV4 | 2 | 2 | 5 | 1 | 9 |
| SPRY2 | 3 | 1 | 7 | 5 | 2 |
| GJB1 | 4 | 5 | 1 | 3 | 7 |
| PYCARD | 5 | - | - | - | - |
| WNT5A | 6 | - | - | - | - |
| FERMT1 | 7 | - | - | - | 5 |
| NRROS | 8 | - | - | - | - |
| LYZ | 9 | 4 | 2 | 8 | 10 |
| NPAS2 | 10 | 3 | 6 | 7 | 6 |
| RP11-290L1.3 | - | 6 | 9 | 10 | - |
| ITGA6 | - | 7 | 10 | 6 | 3 |
| ID3 | - | 8 | 3 | 4 | - |
| DUSP6 | - | 9 | 8 | 9 | - |
| TNFRSF14 | - | 10 | - | - | - |
| TMEM184A | - | - | - | - | 4 |
| RP11-284F21.10 | - | - | - | - | 8 |

Table 4: Top-10 ranked genes according to each feature importance ranking method for predicting the PD-0325901 drug response.

## C.2 Breast Cancer Subtype Classification

**Response Variable.** We used the PAM50 intrinsic subtype label as a five-class outcome, which are annotated as Luminal A, Luminal B, HER2-enriched, Basal-like, or Normal-like (Parker et al., 2009).

**Data Preprocessing of Gene Expression Data.** We obtained the data from the `TCGAbiolinks` R package. Starting from the original gene expression data with $19,947$ genes, we log-transformed the expression values $(\log(x+1))$ to reduce the impact of highly skewed counts. For each gene, we computed its empirical variance across all tumors and retained the $M = 5000$ most variable genes as candidate features. The selected genes were then standardized to have zero mean and unit variance.

| Gene | RAMPART | RAMP | Baseline | SHAP | Permutation |
|---|---|---|---|---|---|
| ESR1 | 1 | 3 | 5 | 2 | 2 |
| FOXC1 | 2 | 1 | - | 10 | - |
| TPX2 | 3 | - | 8 | 7 | - |
| FOXA1 | 4 | 6 | 1 | 1 | - |
| ASPM | 5 | - | - | - | - |
| MLPH | 6 | 2 | - | - | - |
| CDK1 | 7 | - | - | - | - |
| FOXM1 | 8 | - | - | - | 5 |
| GATA3 | 9 | 7 | 6 | 3 | - |
| UBE2C | 10 | - | - | - | 7 |
| TFF3 | - | 4 | - | - | - |
| TBC1D9 | - | 5 | 2 | 5 | - |
| XBP1 | - | 8 | - | - | - |
| KIF2C | - | 9 | - | - | - |
| PRR15 | - | 10 | - | - | - |
| SPDEF | - | - | 3 | 8 | - |
| MYBL2 | - | - | 4 | 4 | - |
| AGR3 | - | - | 7 | 6 | 4 |
| DEGS2 | - | - | 9 | - | - |
| SIDT1 | - | - | 10 | - | - |
| ANLN | - | - | - | 9 | - |
| PLK1 | - | - | - | - | 1 |
| CCNB1 | - | - | - | - | 3 |
| TTYH1 | - | - | - | - | 6 |
| KPNA2 | - | - | - | - | 8 |
| SLC25A3 | - | - | - | - | 9 |
| MCM2 | - | - | - | - | 10 |

Table 5: Top-10 ranked genes according to each feature importance ranking method for breast cancer subtype classification.

| GO Biological Process | RAMPART | RAMP | SHAP | Permutation |
|---|---|---|---|---|
| Positive regulation of transcription regulatory region DNA binding | 100.0 | 100.0 | 100.0 | – |
| Prostate glandular acinus development | 100.0 | 100.0 | 100.0 | – |
| Prostate gland epithelium morphogenesis | 100.0 | – | 100.0 | – |
| Uterus development | 100.0 | – | 100.0 | – |
| Lung epithelial cell differentiation | – | – | 100.0 | – |
| Regulation of miRNA transcription | 82.32 | 82.32 | 82.32 | – |
| Regulation of epithelial to mesenchymal transition | 59.37 | 59.37 | 59.37 | – |
| Positive regulation of mitotic cell cycle | 49.79 | – | – | – |
| Mitotic cell cycle phase transition | 42.00 | – | – | – |
| Male gonad development | 41.44 | – | – | – |
| Stem cell differentiation | 31.82 | – | – | – |
| Developmental growth | 17.55 | – | – | – |
| Cell division | 15.62 | – | – | – |
| Tube morphogenesis | – | 14.70 | – | – |
| Cell population proliferation | 13.63 | – | – | – |
| Negative regulation of transcription by RNA polymerase II | 10.20 | – | 10.20 | – |
| Positive regulation of transcription by RNA polymerase II | – | – | 9.78 | – |
| Regulation of cell cycle | – | – | 9.59 | – |
| Positive regulation of nucleobase-containing compound metabolic process | 6.16 | – | – | – |
| Positive regulation of macromolecule metabolic process | 5.25 | – | – | – |
| Double-strand break repair via break-induced replication | – | – | – | 100.0 |
| Regulation of DNA-templated DNA replication initiation | – | – | – | 100.0 |
| Positive regulation of mitotic metaphase/anaphase transition | – | – | – | 100.0 |
| DNA replication initiation | – | – | – | 100.0 |
| G2/M transition of mitotic cell cycle | – | – | – | 100.0 |
| Mitotic nuclear division | – | – | – | 33.21 |

Table 6: GO biological processes significantly enriched (FDR $\leq$ 0.05) and fold change (capped at 100) in top-ranked genes across feature selection methods. The baseline top-10 genes had no processes significantly enriched at this level.

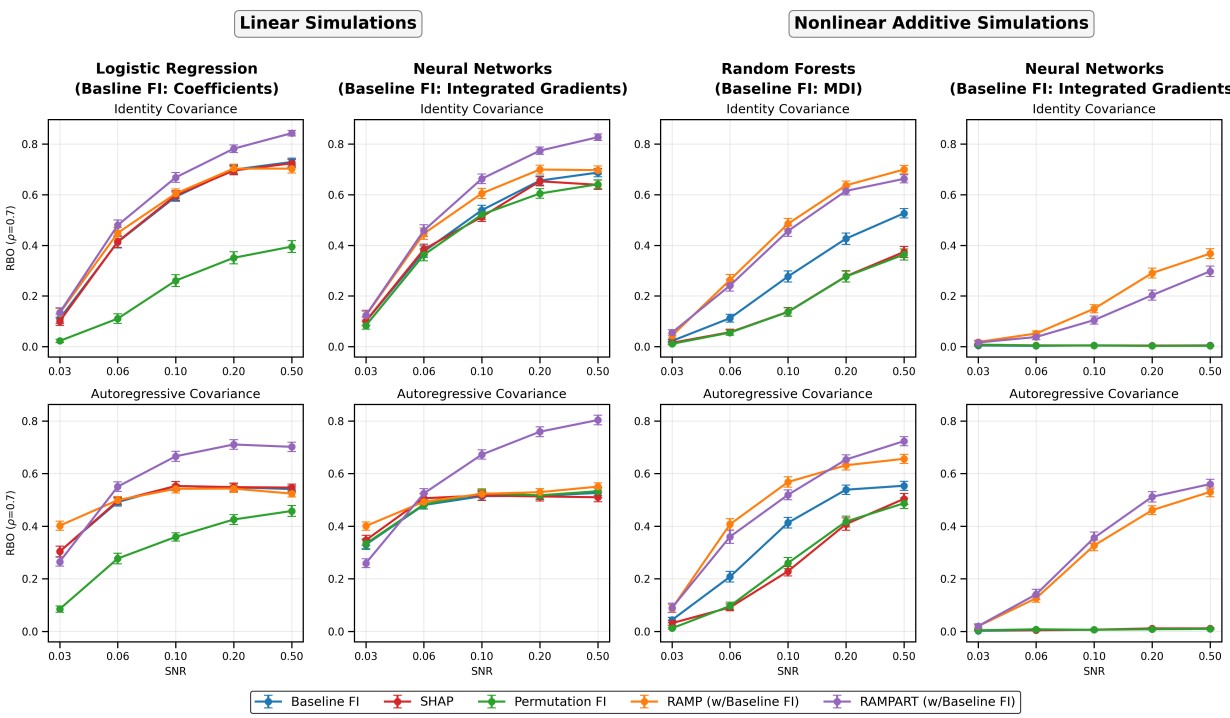

Figure 4: Ranking accuracy (RBO with $\rho = 0.7$) for classification tasks ($M = 1000$)

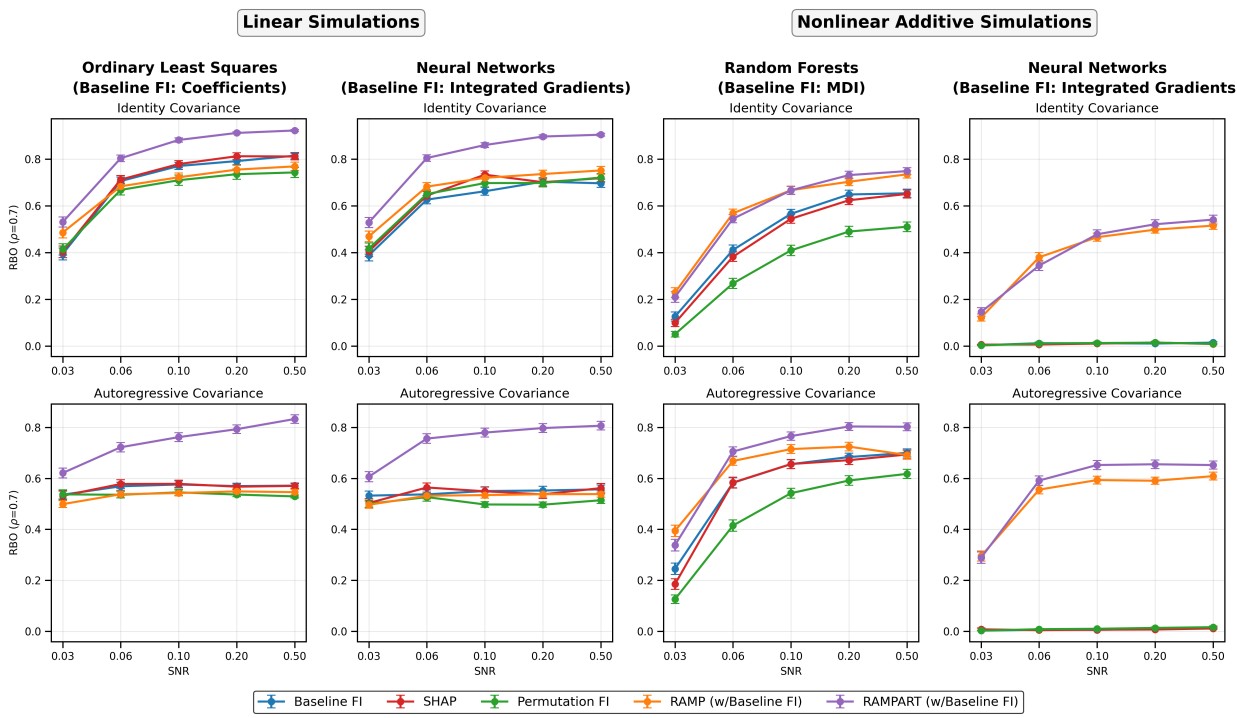

Figure 5: Ranking accuracy (RBO with $\rho = 0.7$) for regression tasks ($M = 1000$)

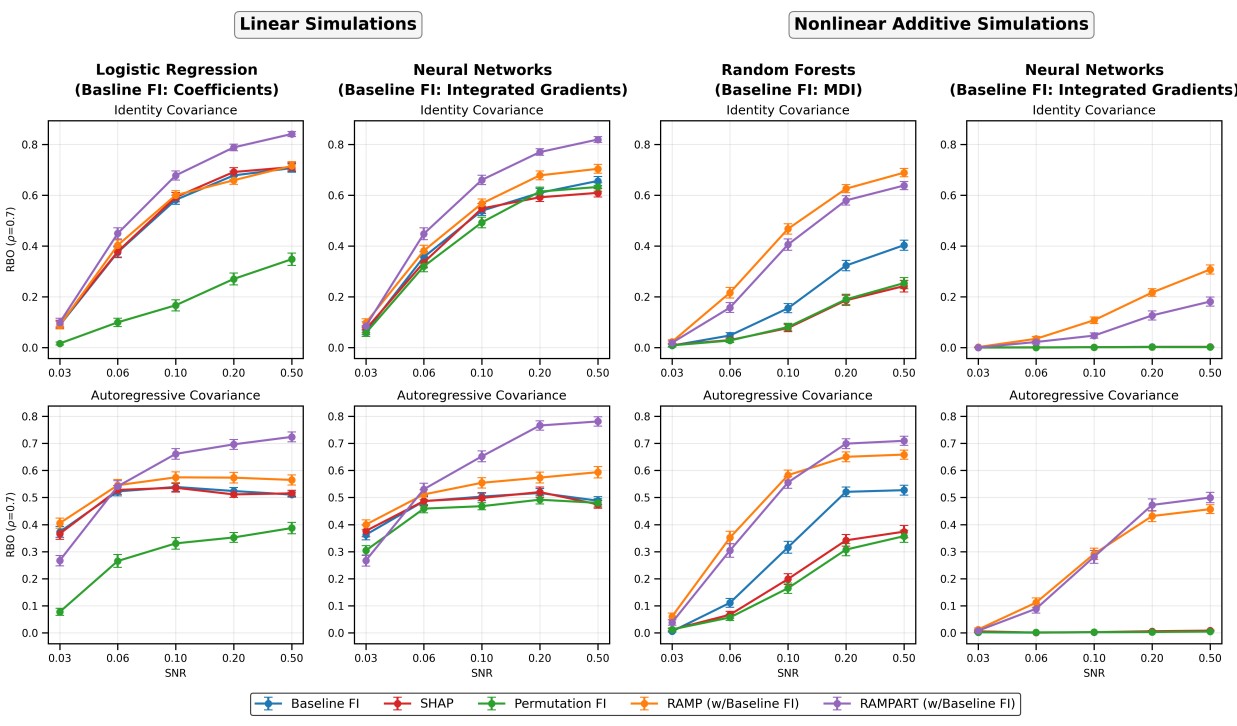

Figure 6: Ranking accuracy (RBO with $\rho = 0.7$) for classification tasks ($M = 2000$)

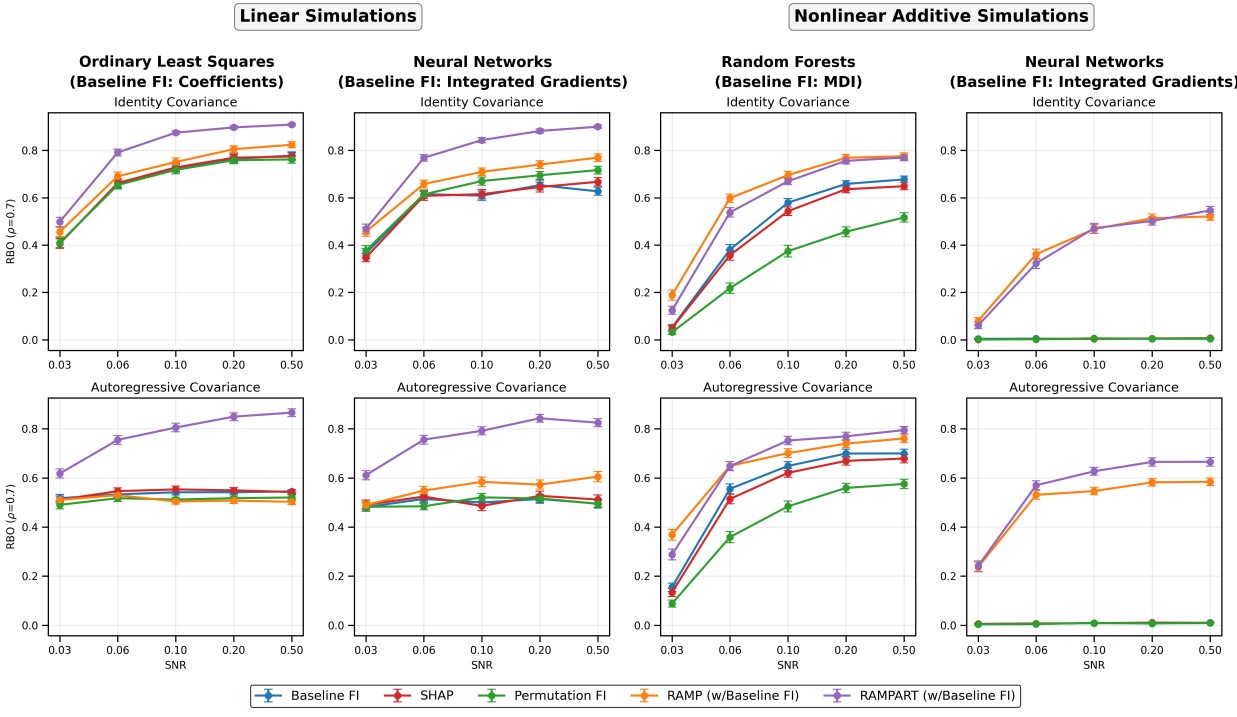

Figure 7: Ranking accuracy (RBO with $\rho = 0.7$) for regression tasks ($M = 2000$)

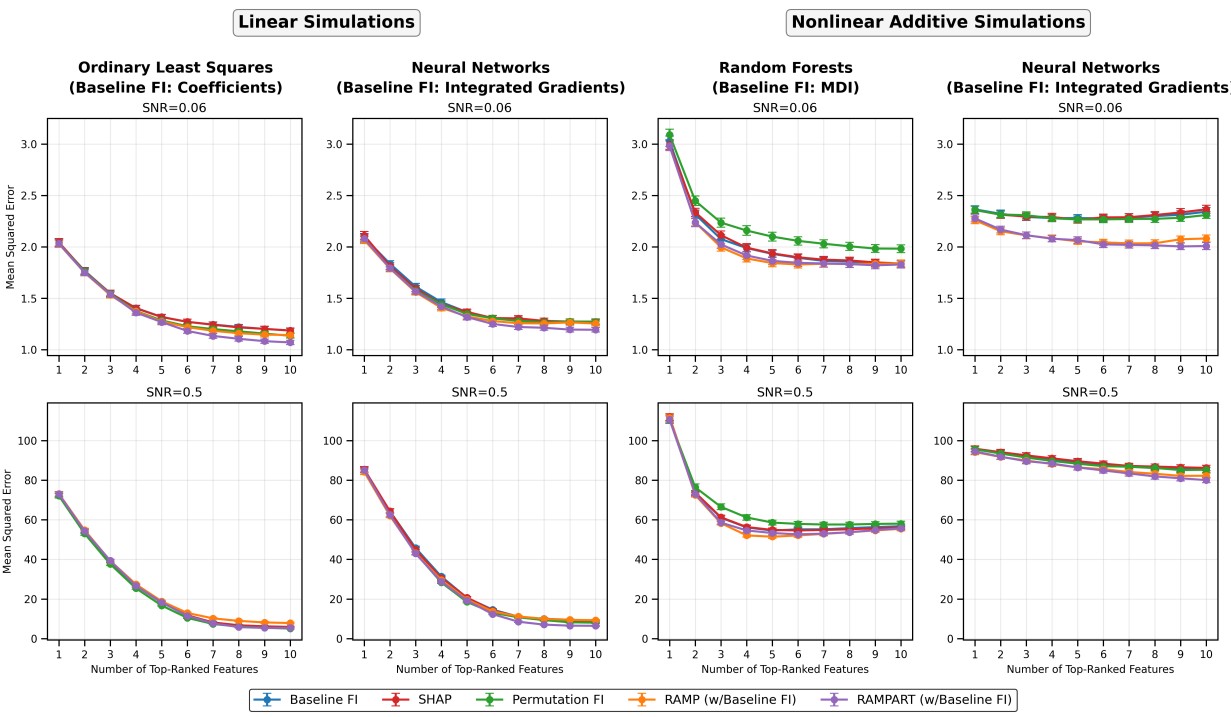

Figure 8: Mean-Squared error vs. number of top-ranked features used as predictors in ablation simulation with identity covariance ($M = 500$)

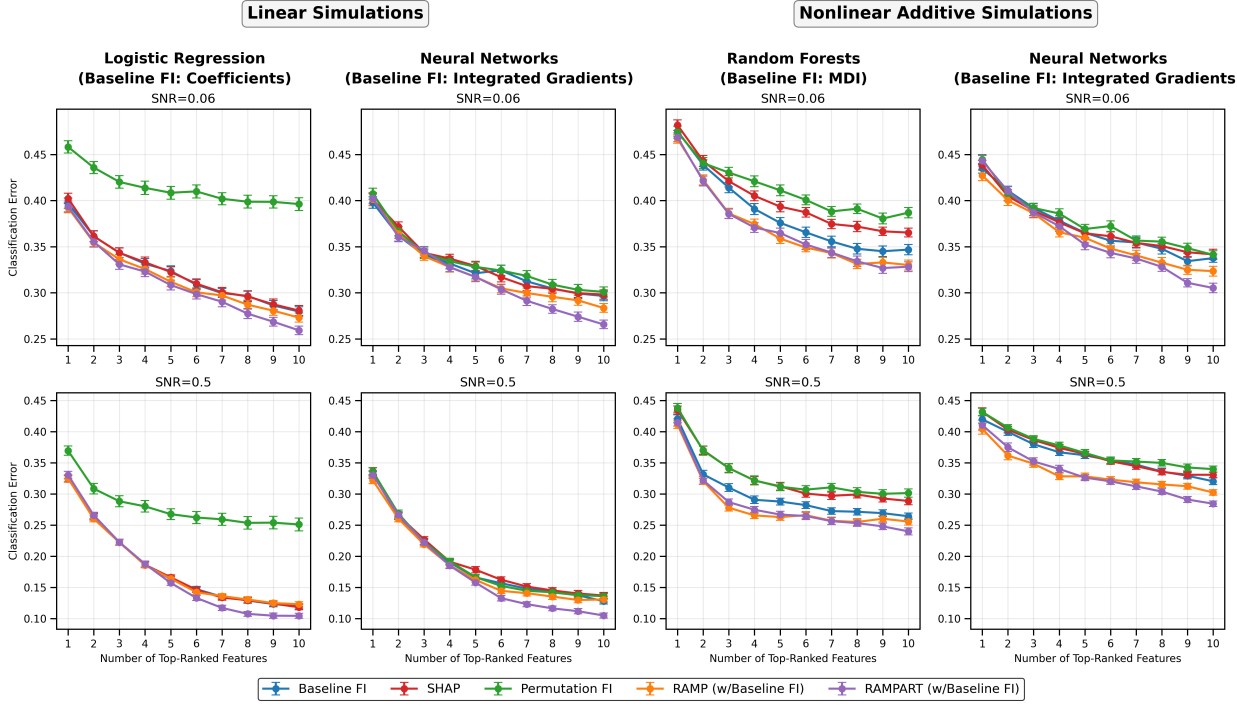

Figure 9: Classification error vs. number of top-ranked features used as predictors in ablation simulation with identity covariance ($M = 1000$)

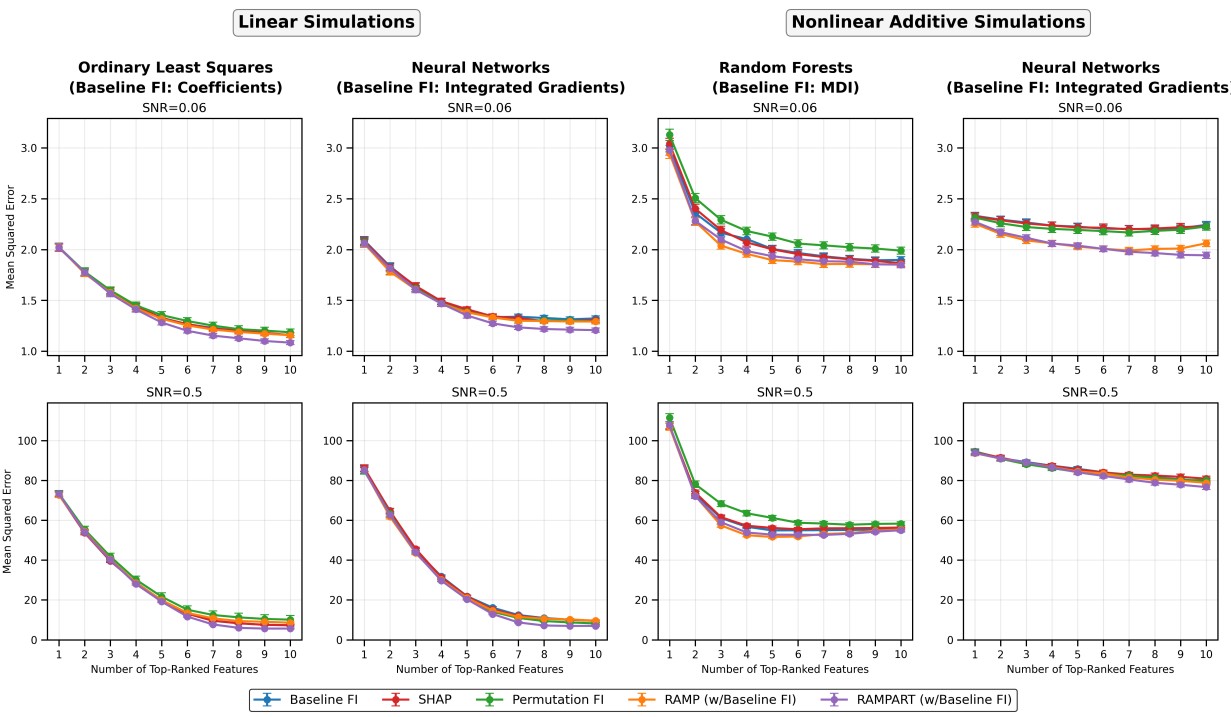

Figure 10: Mean-Squared error vs. number of top-ranked features used as predictors in ablation simulation with identity covariance($M = 1000$)

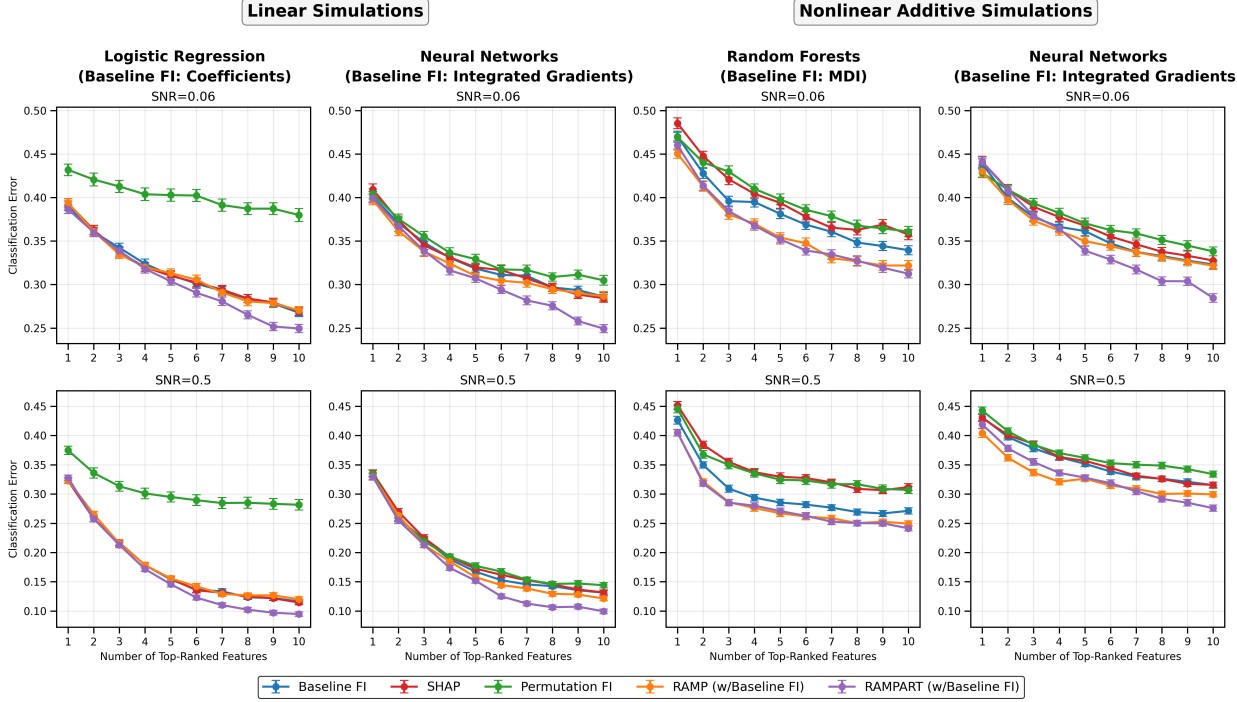

Figure 11: Classification error vs. number of top-ranked features used as predictors in ablation simulation with identity covariance ($M = 2000$)

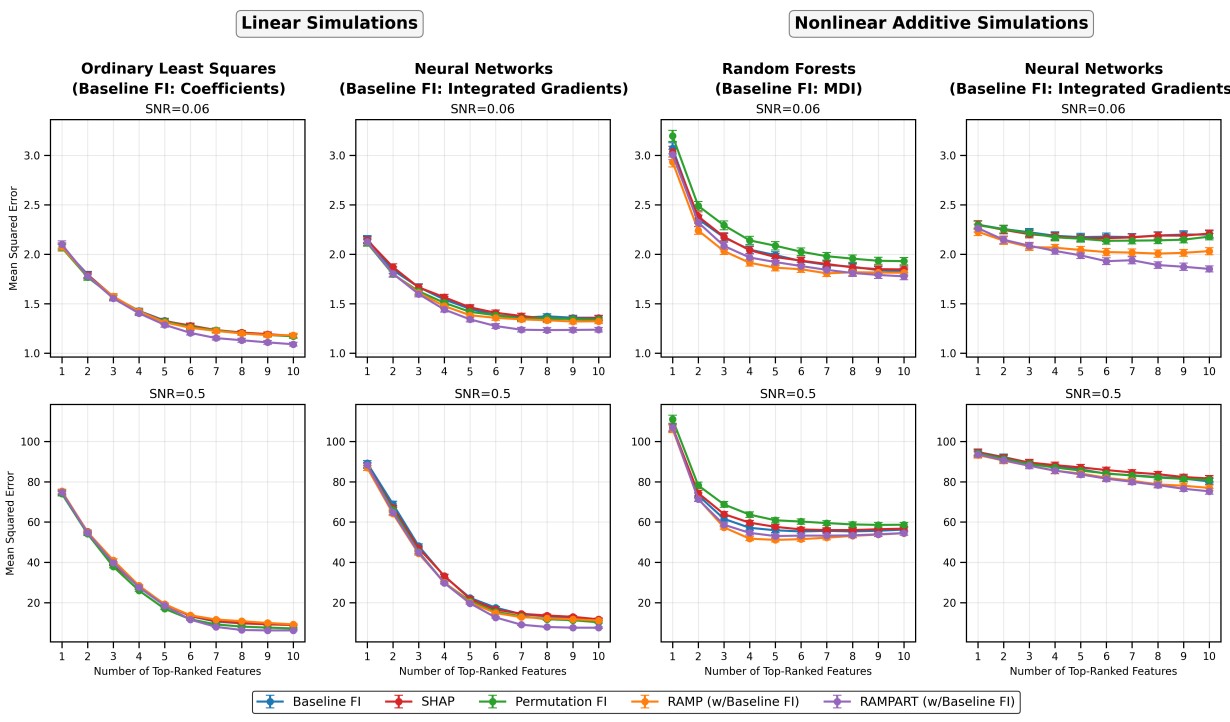

Figure 12: Mean-Squared error vs. number of top-ranked features used as predictors in ablation simulation with identity covariance ($M = 2000$)

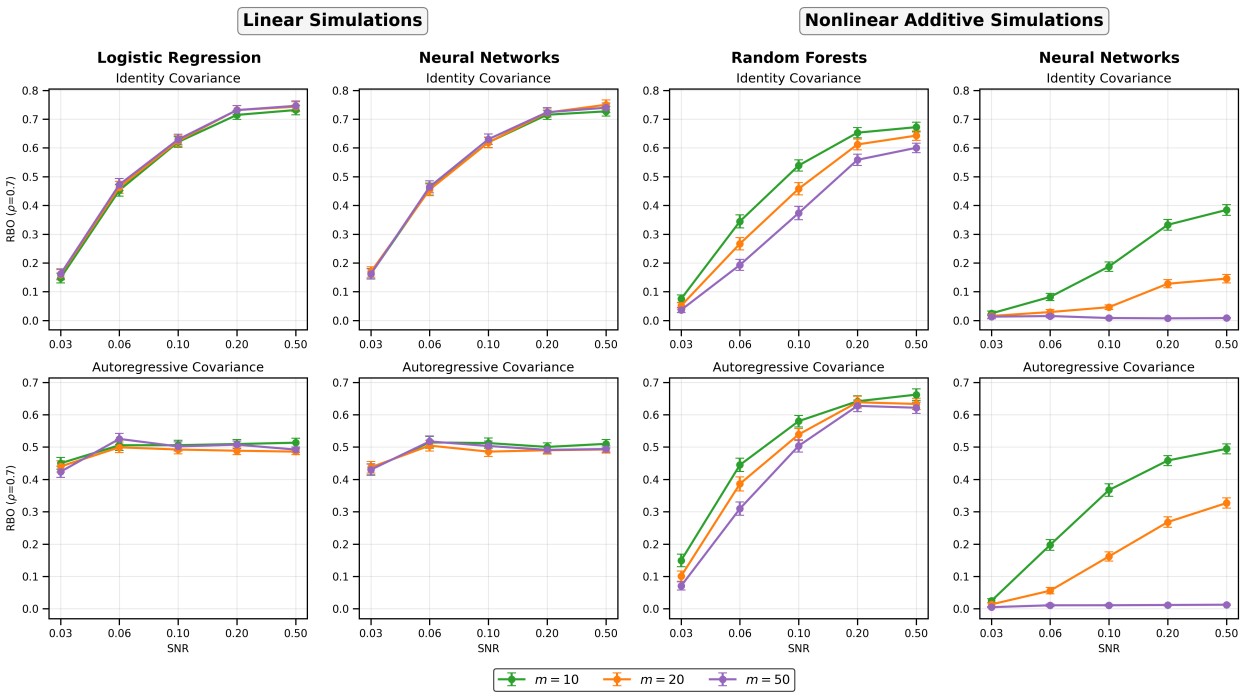

Figure 13: Effect (RBO with $\rho = 0.7$) of minipatch size for classification ($M = 500$)

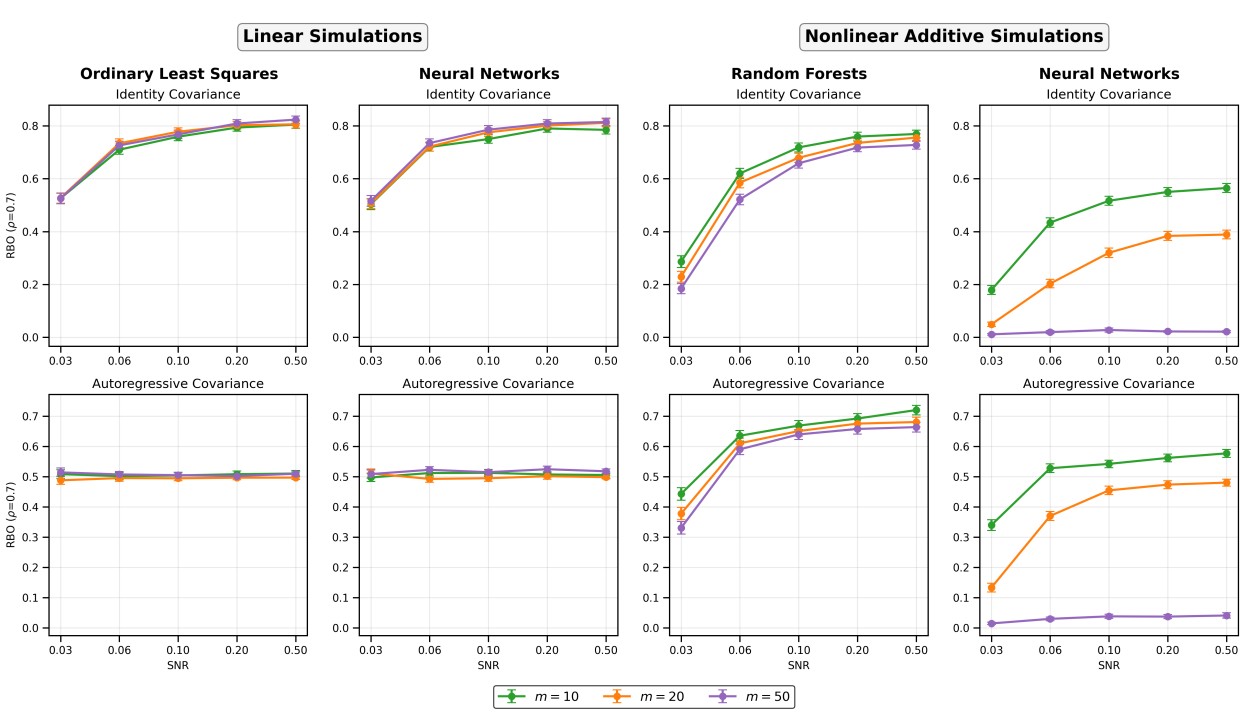

Figure 14: Effect (RBO with $\rho = 0.7$) of minipatch size for regression ($M = 500$)

