# OpenReview forum: "Top-$k$ Feature Importance Ranking"
_TMLR — Accepted by TMLR_

### Review · Reviewer_YJRp · 2025-10-23

**Summary Of Contributions:**

The paper introduces  general framework for top-k feature importance ranking RAMPART.

PRos:
- It identifies the limitation of existing feature importance approaches that only compute scores and sort them post-hoc, emphasizing the need for directly optimizing ranking accuracy for the top-k most important features.
- It proposes a minipatch-based ensembling method that aggregates importance rankings obtained from random subsamples of observations and features to reduce variance and feature correlation bias.
- It extends RAMP into RAMPART, which can narrows the candidate feature set, allocating computational resources adaptively to promising features.
- The experimental section is fairly comprehensive and provides adequate evidence to support the paper’s claims.

Cons:
- The technical motivation could be better articulated in introduction. The introduction moves quickly from the problem to the proposed solution, without clarifying the rationale for adopting this specific technique.
- The paper lacks illustrative examples. Including a few examples would help readers better understand the proposed techniques.
- The discussion section could be expanded to include limitations or potential extensions of the method. Or you should transform the section title into conclusion.

**Audience:**

Yes

**Audience Explanation:**

The topic of feature importance ranking is relevant to researchers in interpretable machine learning and model explainability, which are active areas within TMLR’s audience.

**Claims And Evidence:**

Yes

**Claims Explanation:**

The theoretical analysis is sound, and the experiments are sufficient to support the main claims.

**Requested Changes:**

- The introduction could better explain the reasoning behind the chosen approach. Providing a brief discussion of why this particular framework or design was selected would make the paper more accessible and logically coherent.
- Consider adding one or two concrete examples to help readers intuitively understand how the proposed method works and what specific problems it addresses.
- The discussion could be expanded to include key limitations or possible extensions of the method. Alternatively, if no further analysis is added, it may be clearer to rename this part as a conclusion section.

---

> ### Author Response · Authors · 2025-11-24
>
> We thank the reviewer for the insightful and constructive review, and for the careful balance between highlighting the strengths of the paper and suggesting concrete ways to improve its exposition. In revising the manuscript, we focused on strengthening the technical motivation, adding concrete examples, and expanding the discussion of limitations and extensions, as you recommended.
>
> First, the reviewer observed that the original introduction moved quickly from the problem statement to the proposed solution, without fully explaining why the specific combination of minipatches and recursive trimming is a natural response to the limitations of existing top-$k$ feature-ranking pipelines. In the revised introduction we now sharpen this motivation: after describing the challenges of naively computing importances for all features and then sorting them, we emphasize that our goal is to target the top-$k$ ranking task directly, which leads us to a model-agnostic wrapper perspective. Rather than proposing a new attribution method, we treat existing importance procedures as black-box primitives and design a ranking algorithm that uses inexpensive subsampled fits to screen many features while concentrating computation on the most promising candidates. We have additionally revised the Contributions section to more directly connect to the stated challenges to further improve clarity.
>
> Second, the reviewer recommended including concrete examples to help readers understand how the framework works and where it is most useful. In response, both the introduction and Section 2.2 now contain additional examples illustrating scenarios where a few latent drivers are represented by many correlated proxies. We use this to explain how standard feature-importance pipelines can produce unstable rankings over overlapping proxies and to highlight that the regime of "few true drivers, many correlated proxies, nontrivial attribution cost" is exactly where RAMPART is designed to operate. Sections 2.2 and 2.3 now also contains further clarification on exactly how the RAMP/RAMPART works with the black-box model $\mathcal{M}$. In the empirical studies section, we have also expanded from a single drug-response case study to two genomics applications, adding the TCGA breast-cancer subtype study in Section 4.2.2 and Appendix C.2 as a detailed real-data example of how RAMPART produces an informative, biologically coherent gene panel.
>
> Third, the reviewer suggested that the discussion section should more clearly articulate limitations and possible extensions. We have therefore substantially revised Section 5 into a genuine discussion. It now explicitly acknowledges that RAMPART is a wrapper method that may inherit assumptions and biases from the underlying attribution procedure, and that our current guarantees rely on stylized separation and consistency conditions that may be difficult to verify in practice. We also outline several directions for future work, including better handling of strong feature correlation, adaptive minipatch sizes and sampling schemes, extensions beyond tabular settings, and sharpened theory in regimes that mirror high-stakes applications.
>
> We are sincerely thankful to the reviewer for these suggestions, which we have carefully considered in the revision, which have helped us improve the narrative structure and provide a clearer view of the method’s assumptions, limitations, and extensions.

---

### Review · Reviewer_3zmm · 2025-11-09

**Summary Of Contributions:**

This paper introduces RAMPART (Ranked Attributions with MiniPatches And Recursive Trimming), a new framework designed specifically for the task of accurately and efficiently ranking the top-k most important features in a dataset.

The authors argue that traditional methods, which first calculate importance scores for all features and then rank them, are inefficient and unstable in high-dimensional settings. RAMPART addresses this with a two-part strategy. First, it uses an ensembling technique called RAMP, which trains models on random subsamples ("minipatches") of both observations and features to produce more stable rank estimates. Second, it incorporates this into a "recursive trimming" process, which iteratively discards the least promising half of the features, progressively focusing computational resources on the most likely top candidates. Through theoretical analysis, simulations, and a cancer genomics case study, the paper demonstrates that RAMPART outperforms popular feature importance methods like SHAP and permutation importance, providing more accurate and meaningful rankings.

I think one of the key strengths is the strong validation. The authors support their claims with a comprehensive evaluation, including theoretical guarantees for recovering the correct ranking, extensive simulation studies, and a compelling real-world genomics case study where RAMPART's findings were validated with established biological knowledge.

On the other hand, it's still dependent on base importance metric. RAMPART is a meta-algorithm that improves the ranking process for any given feature importance measure. However, its final output is still dependent on the choice of this base metric (e.g., MDI, Integrated Gradients), and it does not resolve the fundamental debate over which measure best represents "true" importance.

**Audience:**

Yes

**Audience Explanation:**

I think explainable AI researchers will be interested in this submission. This submission addresses a practical problem of ranking important features, which is a more specific and often more useful task than just assigning importance scores.

**Claims And Evidence:**

Yes

**Claims Explanation:**

I think this paper mainly has two claims:
1) RAMPART is more accurate for top-k feature ranking than existing methods.
2) RAMPART's design (minipatch ensembling + recursive trimming) is what drives its performance.


For 1), this paper provides extensive simulation studies (Section 4.1, Figures 1, 4-7) across a wide variety of settings. In these controlled experiments where the ground truth is known, RAMPART consistently achieves the highest ranking accuracy (measured by Rank-Biased Overlap), especially in more challenging scenarios with correlated features and higher dimensions.


For 2), the authors cleverly isolate the effects of the two components. By comparing RAMPART to RAMP (which uses minipatches but not trimming), the results show that while RAMP improves over baseline methods, RAMPART's adaptive trimming provides an additional performance boost. This demonstrates that the progressive focusing of computational resources is a key contributor to its success.

**Requested Changes:**

Overall, I think this paper is in good shape.

---

> ### Author Response · Authors · 2025-11-24
>
> We thank the reviewer for the very positive and encouraging review, and for the clear and accurate summary of our main ideas. We appreciate that the reviewer found both the theoretical analysis and the experiments convincing, and we have taken the reviewer's comments into account when revising the manuscript.
>
> The reviewer emphasizes that RAMPART remains dependent on the choice of base importance metric (e.g., MDI, SHAP, Integrated Gradients) and that our contribution should be understood as a meta-algorithm that improves top-$k$ ranking given such a metric, rather than as a solution to the question of which attribution definition best represents "true" feature importance. We agree with this perspective and, in light of this comment, we have made this viewpoint more explicit. In the introduction and contribution summary we now describe our approach as a complementary, model-agnostic framework in which existing importance procedures are treated as black-box primitives and wrapped in a ranking algorithm that uses subsampled fits and recursive trimming to target the top-$k$ task directly. In the Discussion (Section 5) we further note that, because RAMPART is a wrapper method, it may inherit the modeling assumptions and potential biases of the underlying attribution procedure, and that our guarantees are expressed in terms of properties of this base mechanism. These additions are intended to make clear that our work does not attempt to resolve the broader attribution-definition debate, but instead offers an efficient and statistically grounded ranking strategy for whichever notion of importance a practitioner deems appropriate.
>
> We are grateful to the reviewer for this clarifying comment and for the overall positive assessment, both of which we have carefully considered and incorporated into the revision.

---

### Review · Reviewer_GoeQ · 2025-11-14

**Summary Of Contributions:**

This paper introduces RAMPAR, a novel framework for accurately ranking the top-k most important input features. The method combines a minipatch ensemble approach with an adaptive sequential halving strategy to focus computational resources. The authors provide rigorous theoretical guarantees and demonstrate the method's superiority through simulations and a case study using a cancer genomics dataset.

Strengths:

1.	The method is clearly designed for the specific goal of top-k ranking, effectively leveraging the task's inherent properties. By combining ensembling and adaptive resource allocation, the framework is demonstrably more robust and yields consistently better performance than standard feature importance methods across diverse simulation settings.

2.	The article contains substantial and rigorous theoretical derivation and analysis. The authors provide explicit sample complexity bounds and demonstrate that RAMPART achieves the correct ranking with a higher probability than RAMP while maintaining the same order of total computational complexity.

3.	The experimental results, particularly the Rank-Biased Overlap (RBO) scores in the simulations and the biological validation in the genomics case study, effectively demonstrate the method's superior performance compared to strong baselines.

Weaknesses:

1.	The theoretical foundation relies on several data assumptions, such as Unbiased Ordering (Assumption 3), which may not hold in real-world scenarios characterized by complex, non-additive feature interactions (e.g., features that only contribute when combined). Given that the practical generalization is only tested on a single high-dimensional genomics case study, the method's broad applicability and stability across diverse real-world datasets remain uncertain.

2.	The paper strongly argues that RAMPART‘s recursive trimming optimizes resource usage by avoiding the costly estimation of low-rank features. However, the reported runtime in the Section 4.2 case study does not fully support this claim. Traditional methods like Random Forest (Baseline FI) and TreeSHAP are executed in 6–13 seconds, which is faster than the RAMPART method (18–19 seconds). This discrepancy casts doubt on the practical significance of the claimed efficiency gains associated with avoiding low-rank feature calculations.

**Audience:**

Yes

**Audience Explanation:**

The topic is important, and the proposed method is interesting and supported by rigorous theoretical analysis.

**Claims And Evidence:**

Yes

**Claims Explanation:**

See my strengths of summary.

**Requested Changes:**

More explanation of the resource consumption is needed. Additional experiments on more practical datasets would strengthen the generalization of the proposed method.

---

> ### Author Response · Authors · 2025-11-24
>
> We thank the reviewer for the detailed and encouraging assessment of our work, and for the clear summary of its contributions and limitations.
>
> Regarding the assumptions, the reviewer noted that our analysis relies on conditions such as Unbiased Ordering (Assumption 3), which may not hold exactly in complex real-world settings. In the revised manuscript we now discuss these assumptions and their limitations more explicitly in the Discussion (Section 5). We emphasize that RAMPART is a wrapper around a base attribution method and therefore may inherit its modeling assumptions and potential biases. We also highlight that refining and empirically probing these conditions is an important direction for future work to address the reviewer's concern that the connection between our theory and realistic applications be stated transparently.
>
> As for additional experiments on more practical datasets, we have broadened the empirical section beyond the original single genomics case study. In addition to the PD-0325901 drug-response analysis, Section 4.2.2 and Appendix C.2 now present a second high-dimensional multi-class classification problem based on the TCGA-BRCA cohort, where the goal is to predict PAM50 breast cancer subtypes from RNA-seq gene-expression profiles. In this new case study, we again compare RAMPART, RAMP, a random-forest baseline, TreeSHAP, and permutation importance under a common pipeline. Our analyses show that more biological processes are significantly enriched by RAMPART's top-ranked genes compared to any of the competing feature-ranking methods.
>
> The reviewer also raised a concern about runtime and resource consumption. In the original submission, the reported wall-clock times for the PD-0325901 case study did not, by themselves, fully support our qualitative claim that recursive trimming improves computational efficiency, since RAMPART was modestly slower than a single random forest or TreeSHAP on that dataset. We have reconsidered how we present these results and have added a dedicated subsection, Section 4.3 ("Runtime and Scalability"), together with a new Table 3 for the revision. This table reports wall-clock times for all methods on both genomics case studies under a common setup, and the accompanying text separates constant-factor overheads on moderate-sized problems from asymptotic scaling with the feature dimension. We explain that, on the drug-response dataset with $M=1104$ genes, RAMPART incurs a modest overhead while providing a more stable top-$k$ ranking, and that when we increase the dimension to $M=5000$ in the TCGA breast-cancer task, RAMPART and RAMP remain well under a minute whereas TreeSHAP and permutation importance slow down more noticeably. Furthermore, we now emphasize that RAMPART's main advantage lies in reallocating a fixed minipatch budget toward promising features and achieving better scaling in $M$, rather than in being uniformly faster than all baselines in absolute time.
>
> We thank the reviewer again for these thoughtful comments, which we have carefully considered in revising the manuscript and have helped us improve the clarity and completeness of the work.

---

### Author Response · Authors · 2025-11-24

We sincerely thank the action editor and all three reviewers for their thoughtful and constructive feedback. We have carefully considered every suggestion in revising the manuscript, and we have incorporated the requested changes in the revised version, which has been uploaded to OpenReview.

---

### Decision · Action_Editor_WxxU · 2025-12-17

**Recommendation:** Accept as is

**Audience:**

Yes

**Audience Explanation:**

Feature ranking is a fundamental problem in machine learning. It is expected that some audience will be interested.

**Claims And Evidence:**

Yes

**Claims Explanation:**

This paper proposes a general framework for ranking features. Given a feature importance measuring subroutine, the framework progressively identifies top features by optimizing ranking accuracy - this is in stark contrast to many prior works where ranking algorithms were serving as ad-hoc tool. The authors provide theoretical guarantees under certain assumptions, and showed the success of the framework on simulated data.

Overall, the paper is clearly written, and both the theoretical and empirical evidence are convincing.